# An extracellular biochemical screen reveals that FLRTs and Unc5s mediate neuronal subtype recognition in the retina

Jasper J Visser[1], Yolanda Cheng[1], Steven C Perry[1], Andrew Benjamin Chastain[1], Bayan Parsa[1], Shatha S Masri[1], Thomas A Ray[2,3], Jeremy N Kay[2,3], Woj M Wojtowicz[1]*

[1]Department of Molecular and Cell Biology, University of California, Berkeley, Berkeley, United States; [2]Department of Neurobiology, Duke University School of Medicine, Durham, United States; [3]Department of Opthalmology, Duke University School of Medicine, Durham, United States

**Abstract** In the inner plexiform layer (IPL) of the mouse retina, ~70 neuronal subtypes organize their neurites into an intricate laminar structure that underlies visual processing. To find recognition proteins involved in lamination, we utilized microarray data from 13 subtypes to identify differentially-expressed extracellular proteins and performed a high-throughput biochemical screen. We identified ~50 previously-unknown receptor-ligand pairs, including new interactions among members of the FLRT and Unc5 families. These proteins show laminar-restricted IPL localization and induce attraction and/or repulsion of retinal neurites in culture, placing them in an ideal position to mediate laminar targeting. Consistent with a repulsive role in arbor lamination, we observed complementary expression patterns for one interaction pair, FLRT2-Unc5C, *in vivo*. Starburst amacrine cells and their synaptic partners, ON-OFF direction-selective ganglion cells, express FLRT2 and are repelled by Unc5C. These data suggest a single molecular mechanism may have been co-opted by synaptic partners to ensure joint laminar restriction.

*For correspondence: woj. wojtowicz@gmail.com

Competing interests: The authors declare that no competing interests exist.

## Introduction

In many regions of the nervous system, neurons and their arbors are organized in parallel layers. This organization provides an architectural framework that facilitates the assembly of neural circuits in a stereotyped fashion, a crucial feature that underlies function of the structure. Laminated structures are composed of multiple different classes and subtypes of neurons that form distinct connections in specific stratified layers. During development, the cell bodies and/or neurites of these different neuronal subtypes become restricted to one or more distinct strata. Costratification of arbors promotes synaptic specificity by placing appropriate synaptic partners in close proximity to one another. As such, understanding how lamination occurs is essential to uncovering the molecular basis of how highly-specific neural circuits form.

The mouse retina is an excellent system to study lamination. The inner plexiform layer (IPL) of the retina is a stratified neuropil composed of axons and dendrites belonging to ~70 different subtypes of neurons. These neurons synapse selectively on specific partners, forming a complex set of parallel circuits, so a high degree of specificity is required during the wiring process (for review see *Sanes and Zipursky, 2010*; *Hoon et al., 2014*). The IPL has been well-characterized structurally and functionally. Three major class of neurons (bipolar, amacrine, and retinal ganglion cells (RGCs)) form connections with each other in five IPL synaptic sublayers, termed S1-S5 (*Figure 1B*). Most neurons project selectively to just one or a few of these sublayers. There are many genetic and cell biological

**eLife digest** A nervous system comprises complex circuits of neurons connected by junctions called synapses. These connections need to develop in a highly specific manner, which means neurons need to be able to recognize one another and 'figure out' with which neighboring neuron or neurons they should form connections. Neurons do this by physically interacting with one another via proteins on their cell surfaces; these proteins essentially provide instructions to each of the neurons. However, for most neurons, details remain unclear about how they recognize and 'talk' to one another to form the connections needed to develop into working neural circuits.

To form precise connections, neurons must navigate their way to the appropriate location that places them close to the other neurons with which they need to connect (also known as their "synaptic partners"). In many regions of the nervous system, neurons become organized in parallel layers during development such that synaptic partners reside within the same layer. This process is called lamination and it occurs in the retina in the back of the mammalian eye.

Now Visser et al. have searched for the cell surface proteins that are involved in lamination in the mouse retina. This search involved a number of different gene expression, biochemistry and cell biology-based techniques. Visser et al. identified two families of proteins that might control the lamination of many different subtypes of neurons. The findings reveal some of the molecular mechanisms that underlie the formation of neural circuits in the developing retina and suggest that a pair of synaptic partners may use the same recognition proteins to ensure that they target to the same layer. The next step will be to confirm whether the proteins identified are indeed responsible for organizing neurons into distinct layers during the development of the mouse retina.

tools available to study neurons with lamina-specific projections and retinal neurons are amenable to culture *ex vivo* allowing in-depth analysis of the receptor-ligand interactions that underlie laminar organization. For all these reasons we chose the IPL region of the mouse retina as a model system to study lamination.

Extracellular interactions between neighboring neurons or between neurons and their environment mediate molecular recognition events that direct laminar organization by providing instructions to neurons regarding where to grow (through attraction or repulsion), how to organize neurites and with whom to form synaptic connections (for review see *Tessier-Lavigne and Goodman, 1996*; *Kolodkin and Tessier-Lavigne, 2011*; *Lefebvre et al., 2015*). In this way, molecular recognition specificity (i.e. receptor-ligand interactions) translates into wiring specificity. To date, only a small number of interacting proteins and the instructions they provide to neurites during laminar organization of the mouse IPL has been identified (*Matsuoka et al., 2011*; *Sun et al., 2013*; *Duan et al., 2014*).

A global understanding of how laminar organization of the ~70 different subtypes develops in the IPL requires four systems-level criteria: 1) knowledge of all the secreted and cell surface proteins present within the developing structure that are available to mediate recognition events; 2) an inclusive description of which of these recognition proteins can engage in receptor-ligand interactions (the 'interactome'); 3) a comprehensive understanding of the functional consequence each interaction has on developing neurites (i.e. attraction or repulsion); and 4) a complete atlas detailing the expression of every ligand and its cognate receptor in each neuronal subtype to know which cells are capable of recognizing and responding to one another. Together these data will provide a platform for understanding the molecular basis of how complex neural circuits form between many different subtypes of neurons within an entire structure.

Here we employed a combination of systems biology approaches to address these four criteria and begin the process of studying IPL lamination on a global level (*Figure 1A*). To address the first criteria, we analyzed microarray data from 13 different subtypes of IPL neurons and selected genes encoding cell surface and secreted proteins that were differentially expressed – these are good candidates for mediating cell-cell recognition across subtypes. To address the second criteria, we used a modified version of a technology we previously developed (*Wojtowicz et al., 2007*) to perform a high-throughput, receptor-ligand biochemical screen that tested every pairwise combination of these candidate recognition proteins for binding. This screen identified ~50 previously-unreported

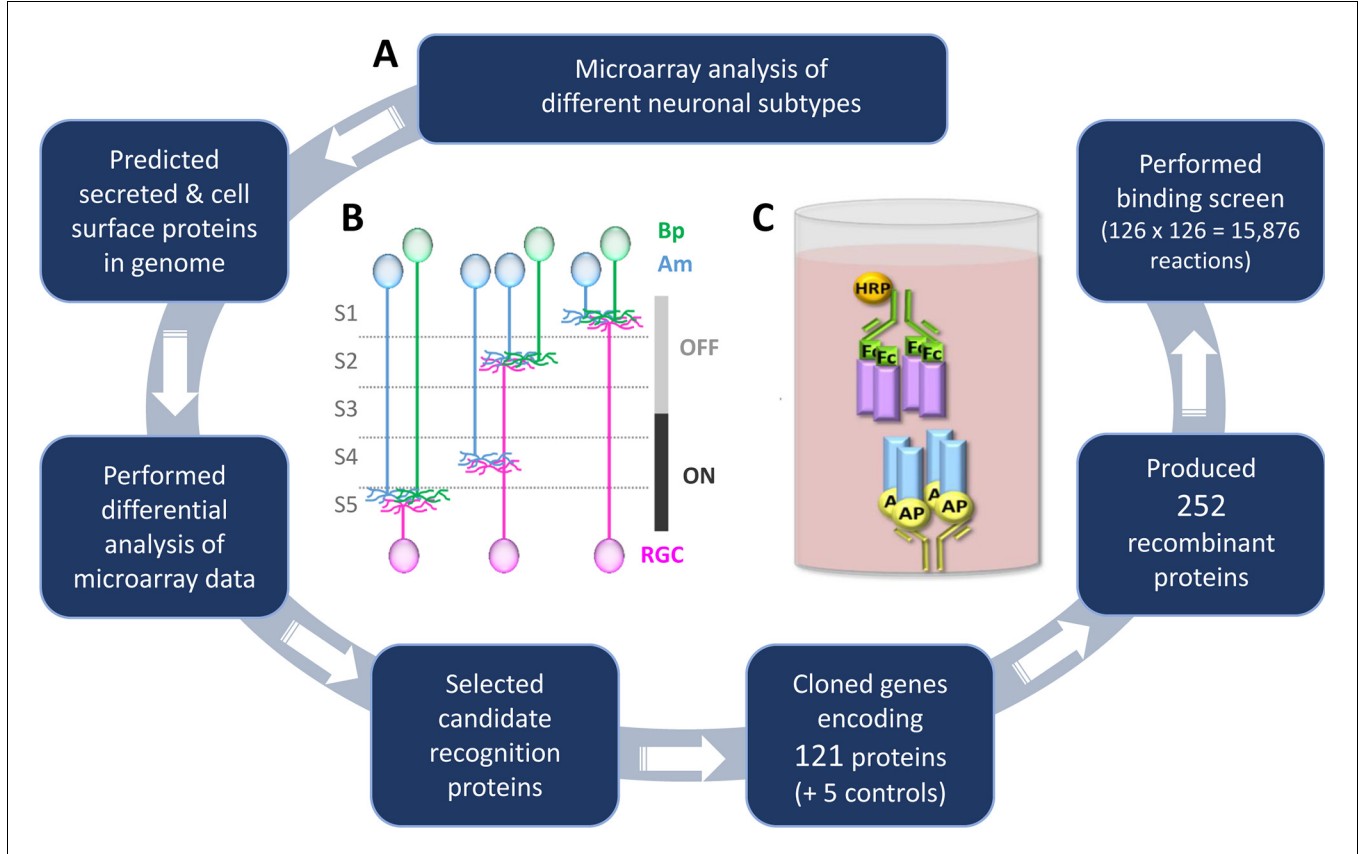

**Figure 1.** Methodology to identify recognition proteins for an extracellular receptor-ligand binding screen. (A) Flow chart describing the process of conducting candidate-based binding screen. A flow chart depicting the process of predicting the cell surface and secreted proteins in the mouse genome prior to candidate selection is outlined in *Figure 1—figure supplement 1*. A table of the 65 candidate genes is included as *Figure 1—source data 1* and a description of the 15 previously-unreported cDNAs that encode new isoforms is presented as *Figure 1—source data 2*. (B) Schematic representation of the IPL showing the five sublayers (S1-S5), three major classes of neurons: amacrines (Am, blue), bipolars (Bp, green), retinal ganglion cells (RGCs, magenta) and the function of the sublayers in visual processing (OFF and ON). Neurite stratifications provide an example of differential laminar organization. (C) Schematic representation of the ELISA-based binding assay. Receptor proteins (blue) tagged with alkaline phosphatase (AP; yellow) are tetramerized on the ELISA plate via an anti-AP antibody (yellow). Binding of tetramerized ligand (purple) tagged with the Fc region of IgG$_1$ (Fc; green) to receptor is detected by inclusion of an anti-Fc antibody conjugated with horseradish peroxidase (HRP; orange).

The following source data and figure supplement are available for figure 1:

**Source data 1.** Table lists the 65 candidate genes selected for the binding screen, the 121 proteins encoded by different isoforms or cleavage products, EntrezGene identifiers and Accession numbers, primer sequences used for cDNA cloning of the extracellular domain, protein type (secreted, GPI-linked or transmembrane) and the protein concentrations for both the AP- and Fc-tagged proteins used in the binding screen.

**Source data 2.** Previously-unreported cDNAs encoding new isoforms.

**Figure supplement 1.** Flow-chart for predicting cell surface and secreted proteins in mouse genome.

receptor-ligand pairs, several between seemingly-unrelated proteins and others between new members within families of proteins previously known to interact.

To investigate whether the receptor-ligand interactions we identified have functional relevance for IPL development, we focused on one family of type I transmembrane receptor-ligand interactions, those between a set of three FLRTs (Fibronectin Leucine-Rich Transmembrane, FLRT1-3) and four Unc5s (Uncoordinated5, Unc5A-D). Some interactions among these molecules have previously been described (*Karaulanov et al., 2009*; *Sollner and Wright, 2009*; *Yamagishi et al., 2011*; *Seiradake et al., 2014*), while others are newly identified in our screen. Members of both the Unc5 and FLRT families exhibit multiple roles in development in a variety of different systems with various

interaction partners (*Bottcher et al., 2004*; *Dakouane-Giudicelli et al., 2014*; *Finci et al., 2015*; *Akita et al., 2015*). Using immunostaining and single cell *ex vivo* stripe assays, we found FLRTs and Unc5s exhibit distinct sublaminar expression patterns in the IPL and elicit repulsion and/or attraction in subsets of retinal neurons. Together these findings are consistent with a role for these families of proteins in mediating differential recognition events between neurons during laminar organization. We propose that, like Contactins, Sidekicks and Dscams in the chick retina (*Yamagata et al., 2002*; *Yamagata and Sanes, 2008*; *Yamagata and Sanes, 2012*), FLRTs and Unc5s are positioned to provide a code for mediating laminar organization in the developing mouse IPL.

## Results

### Identification and production of candidate IPL recognition molecules

Differential expression of extracellular proteins provides a molecular mechanism by which neuronal subtypes distinguish amongst one another. We therefore reasoned that good candidates for mediating neuronal subtype-specific recognition in the IPL are cell surface and secreted proteins that are differentially expressed in different subtypes of amacrine, bipolar and retinal ganglion cells. As no published list of all cell surface and secreted proteins in the mouse genome exists, we first predicted all of the cell surface and secreted proteins using a variety of bioinformatics approaches. A detailed description of this process is outlined in *Figure 1—figure supplement 1*. To identify differentially-expressed recognition proteins (*Figure 1A*), we analyzed microarray data collected from 13 different subtypes of neurons that arborize within different combinations of IPL sublaminae (*Kay et al., 2011b*; *Kay et al., 2012*). The microarray analyses were performed using neurons harvested at P6, a developmental time when extensive neurite extension, arbor refinement, laminar organization and synapse formation are occurring in the IPL.

We identified ~200 genes encoding extracellular proteins that exhibited ≥3-fold difference in microarray expression levels amongst the neuronal subtypes. Based on the domains present in each protein and known players involved in cell-cell recognition, we selected 65 genes as primary candidates and cloned them from retinal cDNA (*Figure 1—source data 1*). Because many of the genes encode more than one protein isoform as a result of alternative splicing or proteolytic cleavage, these primary candidates comprised 121 distinct cDNAs, including 15 splice variants that have not been previously reported (*Figure 1—source data 2*). New splice variants were identified for Ncam1, Netrin5, several Semaphorins and all four Unc5s (i.e. Unc5A-D). The candidate proteins fall into three categories: secreted (26/121; 22%), GPI-linked (17/121; 14%) and type I transmembrane (78/121; 64%). Proteins with multiple transmembranes were not included because their extracellular region is not contiguous and, as such, recombinant protein comprising the entire extracellular domain cannot be readily produced. We cloned the extracellular region of our 121 candidate proteins into two expression plasmids that C-terminally tag the proteins with 1) alkaline phosphatase (AP) or 2) the Fc region of human IgG$_1$ (Fc). Additionally, there is a 6X-His epitope tag on the C-terminus of both AP and Fc.

Recombinant AP- and Fc-tagged proteins were produced by transient transfection of HEK293T cells. As these proteins have a signal peptide but no transmembrane domain or GPI-propeptide, they are secreted into the culture media. For AP-tagged proteins, 106 out of 121 (88%) proteins were produced at optimal concentrations; for Fc-tagged proteins, 110 out of 121 (91%) proteins were produced at optimal concentrations (see *Materials and methods*) (*Figure 1—source data 1* and *Figure 2—figure supplement 1* and *Figure 2—figure supplement 2*). The amount of recombinant protein present in the culture media was quantified using an endpoint kinetic enzymatic assay (AP-tagged proteins) or quantitative Western blots (Fc-tagged proteins) and the levels of protein in the media were normalized. We prefer to use normalized protein concentrations so that the levels of binding can be directly compared between receptor-ligand pairs and interacting pairs with high levels of binding can be identified. However, some proteins were expressed at levels lower than the optimized concentrations (*Figure 1—source data 1*). Nevertheless, these proteins were included in the screen.

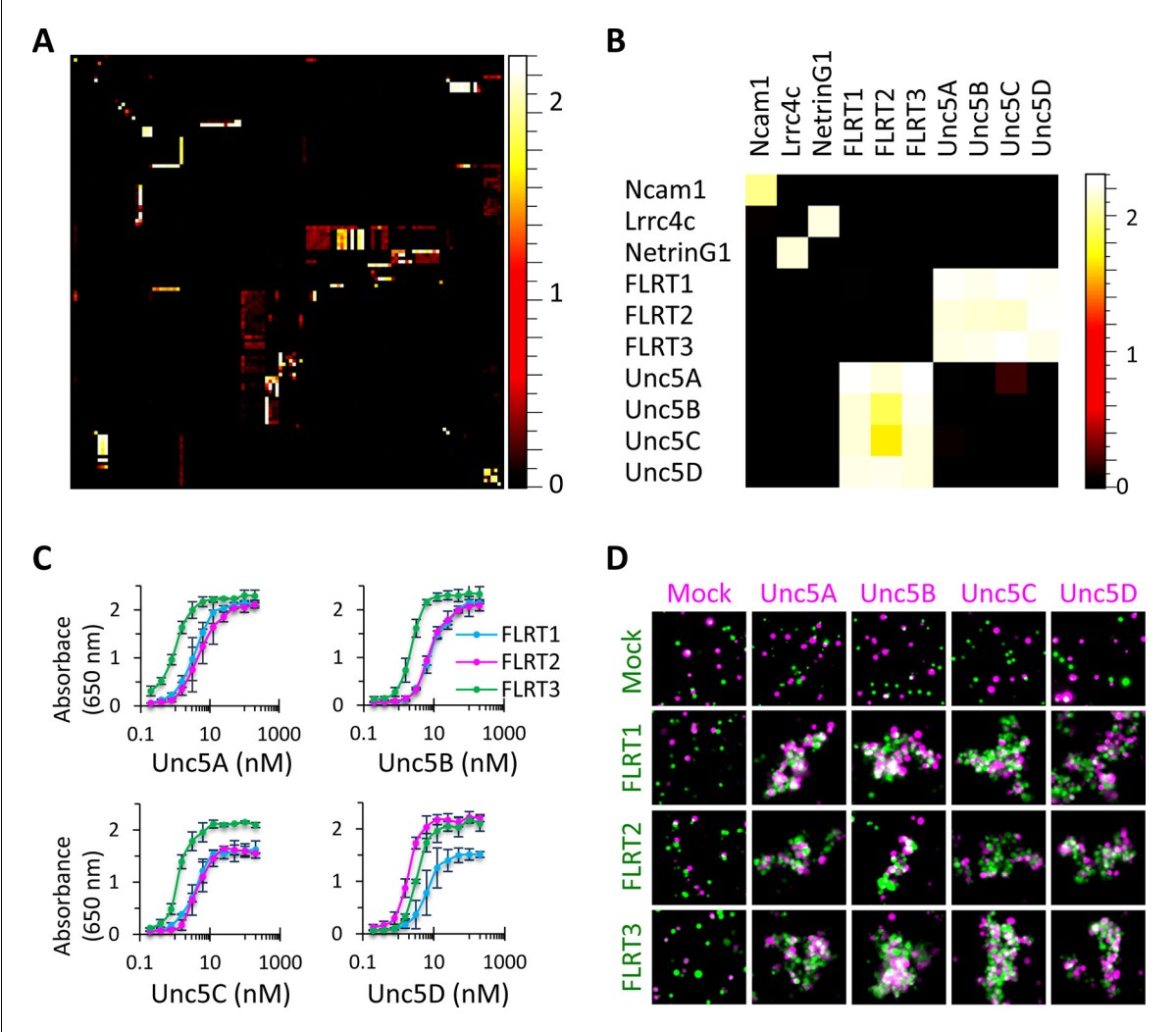

**Figure 2.** High-throughput binding screen results and FLRT-Unc5 interactions. (**A**) 126 x 126 binding matrix. The 126 Fc- and AP-tagged extracellular domain proteins are arrayed along the x and y axes, respectively, in the same order such that homophilic interactions lie on the diagonal. The matrix is colored with a heat map such that high levels of binding are shown in white and no binding is shown in black. Values on the heat map scale represent HRP activity reported as absorbance at 650 nm. Background-subtracted data were deposited in the Dryad database *Visser et al., 2015*. Western blots of the proteins used in the screen are shown in *Figure 2—figure supplement 1* and *Figure 2—figure supplement 2*. (**B**) Subset of binding matrix showing FLRT-Unc5 interactions along with Ncam1 homophilic and Lrrc4c-NetrinG1 heterophilic interactions. Heat maps were generated using Image J (*Schneider et al., 2012*). (**C**) Titration binding curves to monitor FLRT-Unc5 interactions using purified Unc5 protein binding to FLRT attached to an ELISA plate. FLRT1, blue; FLRT2, magenta; FLRT3, green. Three independent experiments were performed in duplicate and average values are plotted. Error bars represent Standard Deviation. (**D**) Cell aggregation assays. CHO.K1 cells expressing full length Unc5 (magenta) and FLRT (green) were mixed together and incubated with shaking. Mixed aggregates of magenta and green cells represent *trans* heterophilic binding. Two independent experiments were performed and representative images are shown.

The following figure supplements are available for figure 2:

**Figure supplement 1.** Western blots of proteins for biochemical screen.

**Figure supplement 2.** Western blots of proteins for biochemical screen.

## Biochemical screen for interactions between candidate recognition molecules

We next screened for interactions between candidate proteins utilizing a high-throughput, extracellular protein ELISA-based binding assay (*Figure 1C*). The screen is a modified version of an assay we

previously described that is quantitative over a 70-fold range (*Wojtowicz et al., 2007*) (see *Materials and methods*). For this study, the workflow was converted from an insect cell strategy to one that would accommodate mammalian proteins. It is largely the case that interactions at the cell surface exhibit low affinities ($K_D \sim \mu M$) and fast dissociation rates (*Vandermerwe and Barclay, 1994*), kinetic properties that allow transient, contact-dependent interactions to occur between recognition proteins expressed on neighboring cells *in vivo* but often make biochemical detection *in vitro* difficult. Our ELISA-based binding assay surmounts this limitation because it utilizes a strategy that tetramerizes the AP-tagged receptor and Fc-tagged ligand proteins (see *Materials and methods*). By inducing tetramers, which provides additive or avidity effects, the assay is highly sensitive allowing proteins with micromolar affinities to be detected at nanomolar concentrations. Such clustering of cell surface proteins (through dimerization, trimerization, tetramerization and pentamerization) is standard practice for detecting ligand-receptor interactions *in vitro* (*Bushell et al., 2008*; *Ramani et al., 2012*; *Ozkan et al., 2013*) as well as in culture experiments where cellular responses to ligands are investigated (*Davis et al., 1994*).

As extracellular interactions are refractory to detection by standard interactome methodologies such as yeast-two-hybrid (*Braun et al., 2009*), our ELISA-based binding assay provided the first platform for performing high-throughput screening of extracellular proteins (*Wojtowicz et al., 2007*). The high-throughput nature of the assay is due, in large part, to the ability to test AP- and Fc-tagged extracellular domain proteins for binding directly in conditioned culture media following transient transfection, thereby obviating the requirement for arduous protein purification. Furthermore, by employing secreted, recombinant proteins, the assay monitors direct protein-protein interactions so it does not suffer the caveat that interactions may reflect indirect binding. As such, this assay, along with two similar, independently-developed ELISA-based binding methods (*Bushell et al., 2008*; *Ozkan et al., 2013*), provides a significant advancement for the study of extracellular protein-protein interactions over low-throughput techniques such as co-immunoprecipitation that, additionally, cannot distinguish between direct and indirect interactions.

To assess which of the 121 candidate recognition proteins can engage in protein-protein interactions as cognate receptor-ligand pairs, we tested them (and five *Drosophila* Dscam1 controls, i.e. 126 proteins) for binding using the ELISA-based assay. The Dscam1 controls were included because some Dscam1-Dscam1 interacting pairs exhibit high levels of binding while others exhibit very low levels, thereby serving as a positive control for the sensitivity of the screen (*Wojtowicz et al., 2007*). We tested the 126 proteins for binding in a matrix which reciprocally tests every pair-wise combination (i.e. 126 x 126 = 15,876 binding reactions) (*Figure 2*; *Visser et al., 2015*). This includes 126 homophilic pairs and 7,875 unique heterophilic pairs. We included reciprocal pairs because sometimes a receptor-ligand interaction will occur in one orientation but not the other. Therefore, by testing each binding pair in both orientations, we decrease our false negative rate.

Interacting proteins identified in the screen were defined as those that exhibited ≥5-fold binding above background levels. Background was determined using absorbance readings at 650 nm ($Abs_{650nm}$) for the 126 control wells that included ligand Fc-tagged culture media (+ anti-Fc-HRP antibody) with mock culture media rather than AP-tagged receptor media (background: mean $Abs_{650nm}$ = 0.064, standard deviation = 0.009). Using this criteria, we identified 192 unique interaction pairs, ~50 of which, to our knowledge, have not been reported in the literature (*Figure 3* and *Figure 4*; *Visser et al., 2015*). To assess the quality of our screen, prior to conducting it we generated a list of 109 receptor-ligand interactions that we expected to see based upon published data. Of these 109 positive control interaction pairs, we identified 91 giving us a false negative rate of 17%. This frequency is lower than published values for the yeast-two-hybrid screen which gives rise to false negative rates between 28 and 51% (*Huang and Bader, 2009*).

## New receptor-ligand pairs identified in the screen

Some of the new receptor-ligand pairs identified involve proteins from families previously not known to associate with one another (e.g. FLRT1-Cntn3, Sema3A-Cntn2 and Ncam1-Dscam) illustrating the importance of conducting unbiased pairwise screens (*Figure 3A-B*). Other new interactions were observed between proteins previously believed to engage exclusively in homophilic, but not heterophilic, binding (e.g. amongst Dscam, Dscaml1 and Sdk2) (*Yamagata and Sanes, 2008*). In addition, new binding pairs were found between members of protein families previously known to interact

with one another (e.g. FLRTs-Unc5s and Dscam-Netrin5) (*Andrews et al., 2008*; *Ly et al., 2008*; *Liu et al., 2009*; *Karaulanov et al., 2009*; *Sollner and Wright, 2009*; *Yamagishi et al., 2011*).

Three of the families included in the screen are the Semaphorins (Sema), Plexins (Plxn) and Neuropilins (Nrp). Previous studies have shown that five classes of Sema ligands (Sema3-7) interact directly with four classes of Plxn receptors (PlxnA-D) or indirectly through binding to the Plxn co-receptors, Nrp1 and Nrp2 (for review see *Yoshida, 2012*; *Gu and Giraudo, 2013*). The specificity of Sema-Plxn interactions is largely restricted within distinct classes (e.g. Sema4s bind PlxnBs and Sema5s bind PlxnAs) with crosstalk occasionally observed (e.g. Sema4C binds PlxnD1). These broadly-defined principles of binding specificity have collectively emerged from a large number of studies that each investigated interactions between limited subsets of Semas and Plxns. Our screen included all members of these families (20 Sema, nine Plxn and two Nrp proteins) and, as such, is the first comprehensive study of Sema-Plxn and Sema-Nrp binding specificity (*Figure 4*). Notably, we observed 1) that Nrp1 and Nrp2 can directly interact with some members of both the Sema4 and Sema6 families; 2) that some Sema3s can interact directly with Plxns in the absence of Nrp1 or Nrp2 (previously only Sema3E was known to interact with PlxnD1 directly and signal in the absence of Nrp) (*Gu et al., 2005*); and 3) new Sema4/5/6-Plxn interaction pairs. In total, we identified twenty-four previously-unreported Sema-Nrp or Sema-Plxn interactions and confirmed four others that had been suggested by genetic interactions (see also *Figure 4—source data 1* and *Figure 4—source data 2*). Together, the results of our screen reveal a wide variety of new interactions among cell surface proteins, which we expect will provide a useful resource to the community of investigators studying cell-cell recognition in a variety of different systems.

## FLRT and Unc5 family interactions

To validate a subset of hits in our screen, we performed additional binding experiments on two families of interacting type I transmembrane proteins, the FLRTs and Unc5s. Interactions between all three FLRT (FLRT1-3) and all four Unc5 (Unc5A-D) family members were observed in the screen; and all pairs exhibited high levels of binding at or near the level of saturation of detection (mean $Abs_{650nm}$ value = 2.14). These families were selected for further study because they were some of the strongest hits, with binding levels comparable to positive controls such as Ncam1 homophilic binding and NetrinG1-Lrrc4c heterophilic binding (*Figure 2B*). Furthermore, of the 12 possible FLRT-Unc5 interactions (i.e. 3 FLRTs x 4 Unc5s), prior to our screen, four had been described in the literature (three in mouse and one in zebrafish) (*Karaulanov et al., 2009*; *Sollner and Wright, 2009*; *Yamagishi et al., 2011*) suggesting that the eight new FLRT-Unc5 binding pairs we identified were likely to represent biologically-relevant interactions rather than false positives.

To test the additional FLRT-Unc5 interactions observed in our screen, we performed titration binding experiments (*Figure 2C*) using purified protein. We utilized a fixed concentration of FLRT receptor on an ELISA plate and varied the concentration of purified Unc5 ligand. In all cases, we observed concentration-dependent binding curves. Because the extracellular region of the proteins used in these titration curves is tetramerized, the FLRT-Unc5 binding constants we observed (i.e. on the order of ~1–10 nM) are much higher than published affinities using monomeric protein in surface plasmon resonance experiments (0.3-21 μM) (*Seiradake et al., 2014*). This observation is similar to findings by Wright and colleagues which showed that pentamerization of extracelluar domains in their ELISA-based binding platform, AVEXIS, can improve the sensitivity of detection over monomeric proteins by at least 250-fold (*Bushell et al., 2008*).

To assess whether all FLRTs and Unc5s can interact between opposing cell surfaces, we performed cell aggregation assays. Full-length versions of FLRT1-3-myc and Unc5A-D-FLAG were co-transfected into CHO.K1 cells along with a plasmid expressing GFP or RFP, respectively. Western blots confirmed that the full-length proteins were produced and immunostaining for the C-terminal epitope tag showed staining around the periphery of the cell consistent with surface expression (data not shown). Using the cell aggregation assay, we tested every combination of FLRTs and Unc5s and found that all pairs interact between opposing cells as evidenced by cell aggregation (*Figure 2D*). By contrast, no clusters were observed between mock transfected cells, FLRT-FLRT or Unc5-Unc5 expressing cells. Together these data confirm that, as observed in our binding screen, *trans* interactions occur between all FLRT-Unc5 pairs.

## FLRTs and Unc5s induce repulsion and attraction in subsets of retinal neurons

We next wanted to know what effect FLRTs and Unc5s have on retinal neuron outgrowth. To investigate the cell biological response of primary retinal neurons (i.e. attraction or repulsion), we performed *ex vivo* stripe assays (*Vielmetter et al., 1990*; *Delamarche et al., 1997*). Because the IPL contains arbors from ~70 different subtypes of neurons, each of which may respond differently (or not at all) to the same protein ligand, it was necessary for us to use a stripe assay that would provide single-cell resolution. The tremendous value of single-cell stripe assays is that they allow the response of an individual subtype of neuron to be observed within a mixed population. As such, we designed and fabricated microfluidic devices (*Figure 5—figure supplement 1* and *Materials and methods*) to pattern 30 μm stripes, a width appropriate for the growth of single IPL neurons whose cell bodies average between 10-30 μm (data not shown). Our design is similar to others that have been used to monitor the effect of a purified ligand on neurite outgrowth of single dissociated neurons (*Weinl et al., 2003*; *Yamagishi et al., 2011*; *Singh et al., 2012*; *Beller et al., 2013*; *Sun et al., 2013*).

We dissected and dissociated neurons from wild-type P6 retinas and cultured individual neurons on FLRT or Unc5 stripes. We reasoned that proteins involved in mediating laminar organization, or other recognition events that play a role in neural circuit formation, would elicit a response (i.e. attraction or repulsion) in only a subpopulation of neurons. While the majority of neurons did not respond to FLRT or Unc5 stripes, growing indiscriminately across them, we observed small populations of neurons (5-18%) that responded to FLRT1 (n=61/375, 16% attractive; n=19/375, 5% repulsive), FLRT2 (n=63/344; 18% repulsive), FLRT3 (n=37/438, 8% attractive; n=33/438, 8% repulsive), Unc5C (n=45/396, 11% repulsive) and Unc5D (n=49/407, 12% repulsive) stripes (*Figure 5A-I*). No significant response of neurons was observed to Unc5B stripes (n=3/380, 1% repulsive) relative to control laminin stripes (n=1/88, 1% repulsive). There also were no attractive or repulsive responses to Unc5A stripes (n=257/257, 100% permissive) but we did observe a modest population-wide reduction in neurite outgrowth and decreased viability (data not shown). Together these data demonstrate that Unc5C, Unc5D, and all three FLRTs mediate recognition events between subtypes of retinal neurons and suggest that FLRTs and Unc5s may contribute to development of the retinal circuit.

## FLRTs and Unc5s exhibit differential expression patterns in the developing IPL

To investigate which subpopulations of retinal neurons are using FLRTs and Unc5s to mediate recognition events involved in wiring, we next assessed the expression of FLRTs and Unc5s in the developing retina using immunostaining of P2, P4 and P6 retinal sections (*Figure 6* and *Figure 6—figure supplement 1*). All FLRT and Unc5 antibodies were highly specific with little to no cross-reactivity as assessed by ELISA using purified protein (*Figure 6—figure supplement 2*). To visualize the boundaries of the five IPL sublaminae (S1-S5), we stained retinal sections with an antibody against vesicular acetylcholine transporter (VAChT). VAChT stains the dendrites of two subtypes of amacrine cells called OFF and ON starburst amacrine cells (SACs) that arborize within functionally-distinct sublaminae S2 and S4, respectively (*Stacy and Wong, 2003*). As such, the positions of the other sublaminae (i.e. S1/3/5) can be inferred relative to the VAChT stain in S2/4 (*Haverkamp and Wassle, 2000*).

At P6 we observed laminar-restricted expression patterns for all FLRTs and three out of the four Unc5s (*Figure 6*). FLRT1 expression was largely restricted to neurites that arborize in S1 (*Figure 6A*), FLRT2 was most highly expressed in S2/4 (*Figure 6B*) and FLRT3 expression was largely restricted to S3 (*Figure 6C*). Unc5A was highly expressed in the cell body layers flanking the IPL and, within the IPL, was expressed in neurites that arborize in S1/2/3/5 (*Figure 6D*), Unc5C was most highly expressed in S1/3/5 (*Figure 6F*) and Unc5D expression was largely restricted to S1/5. Unc5B did not show laminar restriction—it was expressed at low levels uniformly across the IPL (*Figure 6E*).

Comparison of the expression patterns observed at P6 with the patterns observed at P2 and P4 (*Figure 6—figure supplement 2*) demonstrates that laminar-restricted expression of FLRT1-3 and Unc5A,C,D is spatio-temporally regulated. Three patterns of developmental regulation were observed. One subset of proteins, FLRT2 and Unc5C, showed broad expression across the IPL at P2 that gradually became sublamina-restricted by P6. A second group, FLRT1 and FLRT3, showed

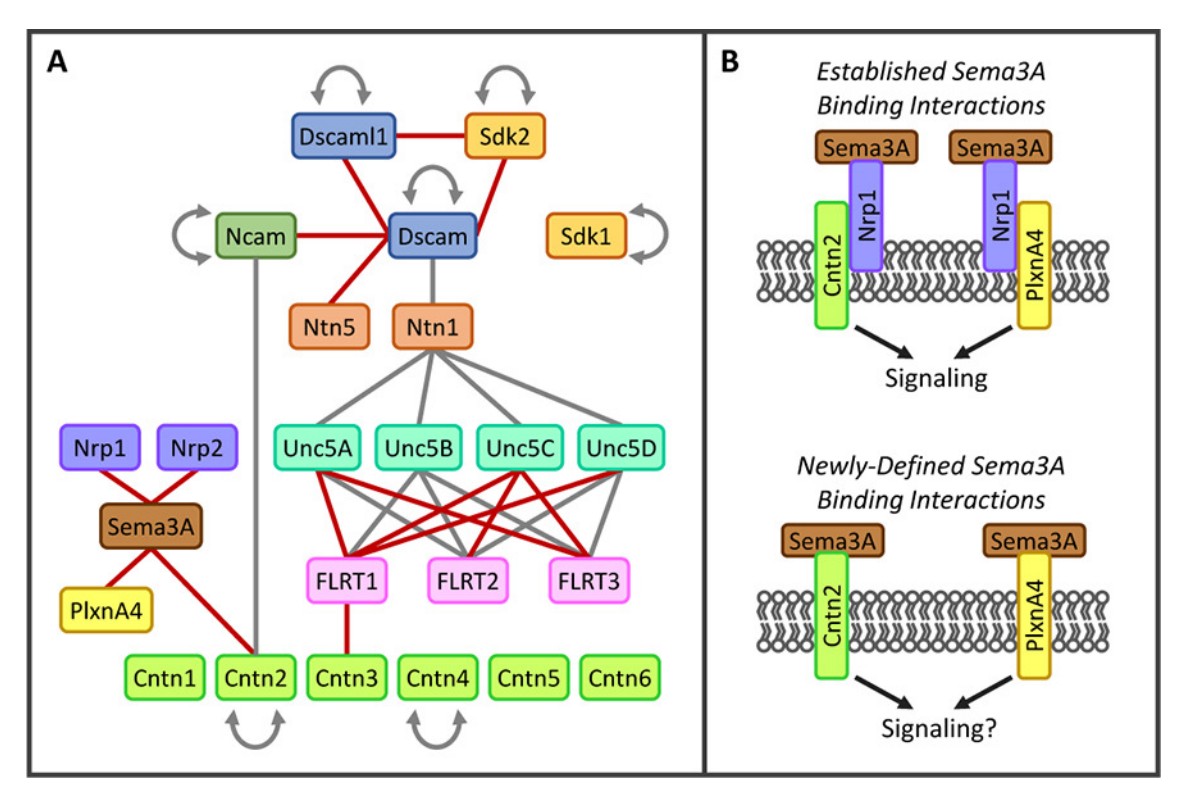

**Figure 3.** New interactions identified in biochemical screen. (**A**) Interactions observed between a subset of proteins included in the screen. Lines indicate direct protein-protein interactions (red line, not previously reported; gray line, previously known). Families of proteins are represented by color. Only one member of the Semaphorin family (Sema3A, brown) and one member of the Plexin family (PlxnA4, yellow) are shown. The complete binding data for all Semaphorins, Plexins and Neuropilins (Nrp, purple) are shown in *Figure 4*. For space considerations, gene names are used for proteins (e.g. Cntn1 for Contactin1). *Figure 1—source data 1* includes full protein names and aliases. (**B**) (*Top panel*) Previous studies have demonstrated that Nrp1 (purple) can form a holoreceptor complex for Sema3A ligand (brown) through *cis* interactions with PlxnA4 (yellow), Cntn2 (green) and a variety of other proteins in the cell membrane (for review see *Yazadani and Terman, 2006*). (*Bottom panel*) Our binding screen identified that Sema3A can engage in direct protein-protein interactions with both PlxnA4 and Cntn2 in the absence of Nrp1.

sublaminar bias already at P2 that changed only slightly as the IPL expanded with age. The final group, Unc5A and Unc5D, added new sublayers at later ages: Unc5A was not observed in the IPL until P6, even though immunoreactivity was detected in neuronal somata at earlier ages, suggesting that IPL innervation by Unc5A-positive cells happens later than other family members. Unc5D, meanwhile, exhibited S1 restriction at P2-4 and then added expression in S5 at P6. Interestingly, the expression pattern of Unc5D may remain dynamic after P6, as immunostaining published by Feldheim and colleagues suggests that, while S5 expression is maintained, S1 expression is lost by P8 (*Sweeney et al., 2014*). The three patterns of laminar restriction we observed – termed 'initially diffuse,' 'initially precise,' and 'stepwise' lamination – have been seen in previous studies of IPL laminar targeting (*Mumm et al., 2006*; *Kim et al., 2010*). The spatio-temporal and laminar-specific expression patterns of the FLRTs and Unc5s suggest that members of both families may contribute to specific cell-cell interactions that mediate these developmental strategies for laminar organization.

## FLRT2-Unc5C cognate ligand-receptor pairs are expressed in repelled neurons

Between P2 and P4, Unc5C and FLRT2 expression patterns become restricted to complementary sublaminae in the IPL with Unc5C concentrated in S1/3/5 and FLRT2 predominantly expressed in S2/ 4 (*Figure 6—figure supplement 2*). Complementary expression suggests that these lamina-specific stratifications may arise due to repulsive interactions between neuronal subtypes expressing FLRT2 and Unc5C. Consistent with this model, our *ex vivo* stripe assays revealed subpopulations of neurons

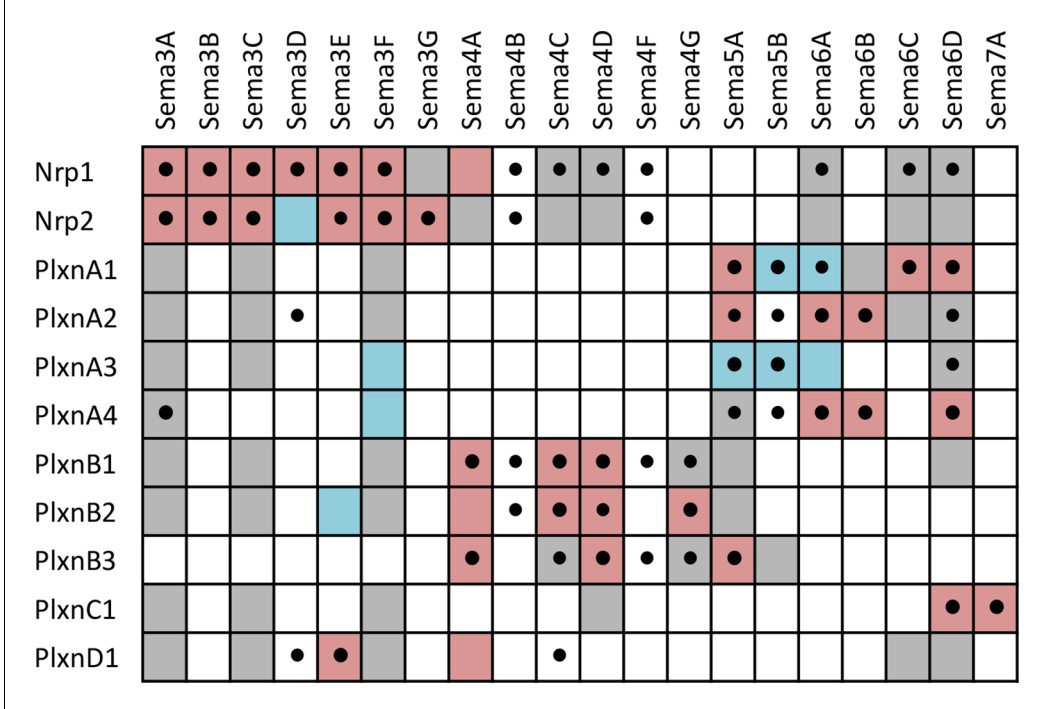

**Figure 4.** Summary of interactions between Sema-Nrp and Sema-Plxn proteins, highlighting new interactions observed in our screen as well as previously known interactions. A complete grid of known interactions was compiled from results reported in ten Semaphorin review articles (*Yazdani and Terman, 2006*; *Neufeld and Kessler, 2008*; *Wannemacher et al., 2011; Hota and Buck, 2012; Neufeld et al., 2012; Yoshida, 2012; Gu and Giraudo, 2013; Roney et al., 2013; Worzfeld and Offermanns, 2014; Masuda and Taniguchi, 2015*) and in independent primary literature searches conducted by several members of our laboratory. We included data from ten review articles because there is considerable variability in the interactions reported (see *Figure 4—source data 1* and *Figure 4—source data 2*). All interactions reported in the reviews were corroborated in the primary literature and are denoted in the table by colored boxes that indicate the type of experiment supporting the interaction. Pink = evidence from cell binding assays, surface plasmon resonance, coimmunoprecipitation, transwell suppression and *ex vivo* explant outgrowth or growth cone collapse. Blue = genetic interactions. Gray, failure to find interaction by one or more of the above methods (i.e. published negative interaction). A black dot (•) indicates a positive interaction observed in our screen. The reference and a description of the supporting data for each previously-known interacting pair are presented in *Figure 4—source data 2*. It is important to note that there are multiple aliases for most *Sema, Plxn* and *Nrp* genes and, as such, our literature searches included these alternative names (e.g. several Sema proteins were initially called collapsins and Sema3B was once called Sema5). These aliases are listed in *Figure 4—source data 3*.

The following source data is available for figure 4:

**Source data 1.** Sema-Nrp and Sema-Plxn interactions published in review articles.
**Source data 2.** Literature search results for Sema-Nrp and Sema-Plexin interactions.
**Source data 3.** Gene name aliases for *Sema, Nrp* and *Plxn*.

that are repelled by FLRT2 and subpopulations of neurons that are repelled by Unc5C (*Figure 5A, D, H*).

We hypothesized that repulsion by Unc5C stripes is due to interactions with FLRT2 expressed on repelled neurons. To investigate this possibility we performed immunostaining on neurons repelled by Unc5C stripes with antibodies against FLRT2 (as well as FLRT1 and FLRT3). Neurons repelled by Unc5C stripes expressed FLRT2 (n=26/26) (*Figure 5J*) but not FLRT1 or FLRT3 (data not shown). Conversely, neurons repelled by FLRT2 stripes expressed Unc5C (n=30/30) (*Figure 5K*). Together these data are consistent with a model wherein interactions between FLRT2 and Unc5C induce mutual repulsion via bidirectional signaling in both the ligand- and receptor-expressing cells.

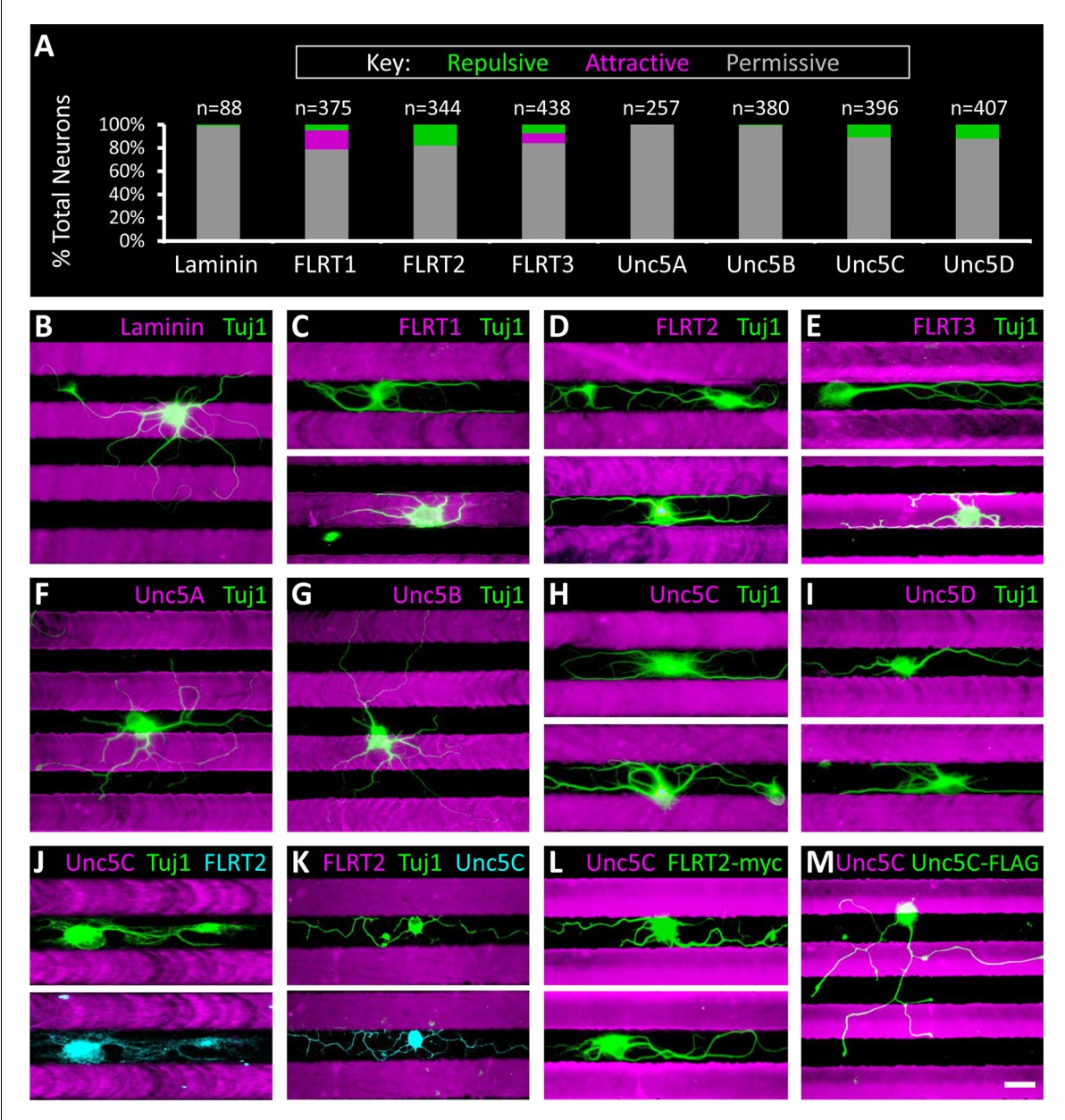

**Figure 5.** Subpopulations of primary retinal neurons respond to FLRT and Unc5 protein in stripe assays. Individual retinal neurons harvested from wild-type retinas at P6 were cultured for 4– 6 days on glass coverslips containing alternating stripes of laminin and a purified candidate recognition protein. (A) Quantification showing the percent of neurons that exhibited a repulsive (green), attractive (magenta) or permissive (gray) response to stripes of the candidate recognition protein. n = total number of neurons scored. Raw data are reported in the main text. (B-I) Example images showing responses of neurons to stripes of the indicated FLRT or Unc5 protein (magenta). Stripes were prepared using microfluidic devices as outlined in *Figure 5—figure supplement 1* and were visualized by addition of BSA-TRITC (magenta) to the purified FLRT or Unc5 protein patterned. As coverslips were coated with the growth-promoting protein, laminin, prior to application of the stripes, the black (unstriped) regions of the coverslip contain laminin. Neurons were immunostained with an antibody against beta-tubulin (Tuj1; green). (J-K) Example neurons co-stained for Tuj1 (green) and FLRT2 (cyan in J) or Unc5C (cyan in K). Neurons that express FLRT2 are repelled by Unc5C stripes (J), while neurons that express Unc5c are repelled by FLRT2 stripes (K). See main text for quantification. (L-M) Gain-of-function stripe assay. Neurons transfected with full-length FLRT2-myc (green) are repelled by Unc5C stripes (L) whereas, neurons transfected with full-length Unc5C-FLAG (green) are not repelled by Unc5C stripes (M). Scale bar, 30 μm.

The following figure supplement is available for figure 5:

**Figure supplement 1.** Microfluidic device design for patterning protein stripes for stripe assay.

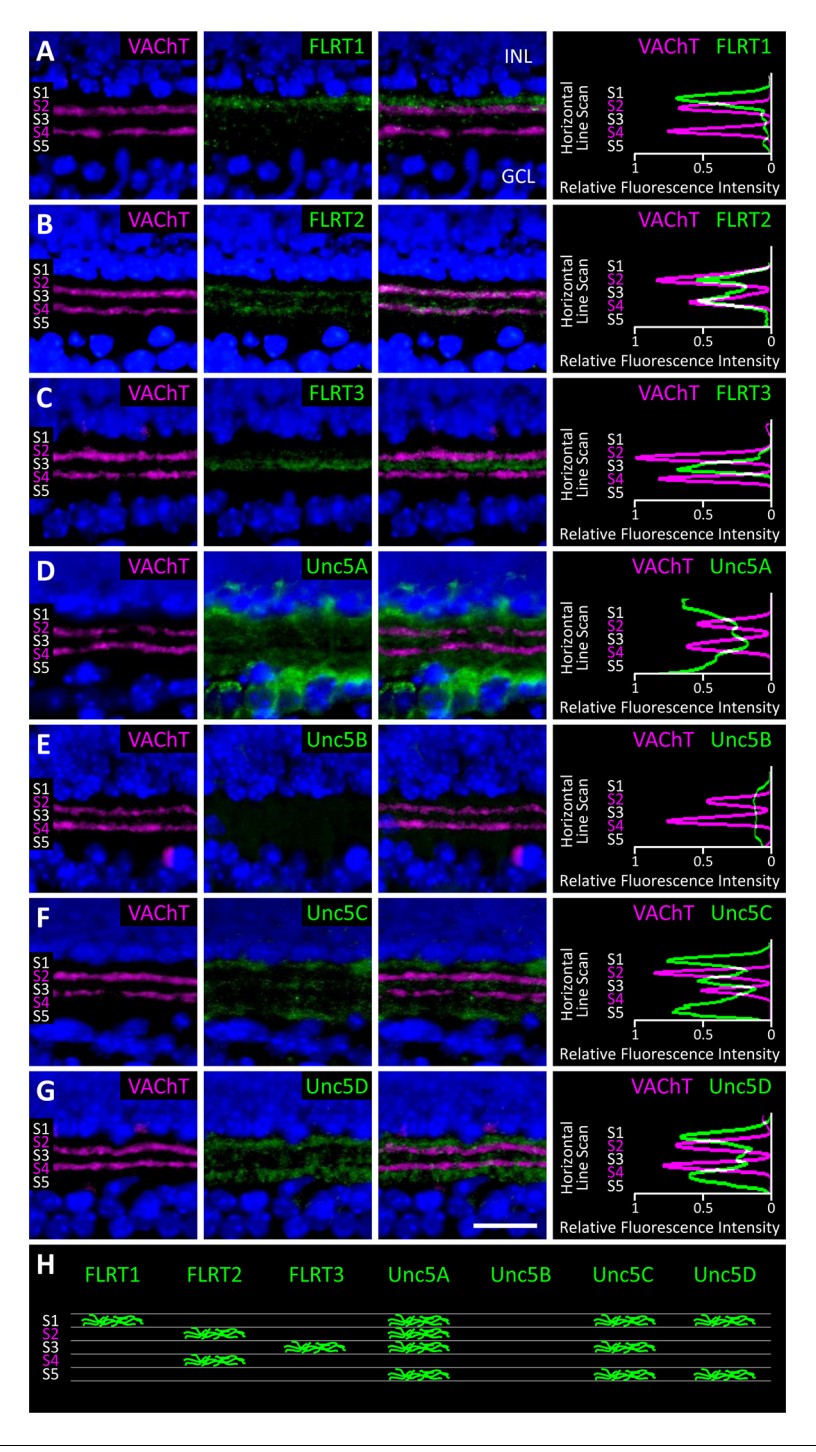

**Figure 6.** Expression of FLRT and Unc5 proteins in the developing IPL. (**A-G**) Retinal sections from C57Bl/6 P6 mice immunostained with an antibody against vesicular acetylcholine transporter (VAChT; magenta), which is expressed by SAC dendrites and thus serves as a marker for sublaminae S2 and S4, and an antibody against one of the FLRTs or Unc5s (green) as indicated in each panel. DAPI (blue) labels cell bodies in the inner nuclear layer (INL) and ganglion cell layer (GCL) flanking the IPL (for schematic see *Figure 1B*). FLRT and Unc5 antibodies were highly specific as demonstrated by ELISA and shown in *Figure 6—figure supplement 1*. Expression patterns at P2 and P4 are shown in *Figure 6—figure supplement 2*. Scale bar, 50 μm. Relative fluorescence of each marker across IPL sublayers S1-S5 is quantified in the histograms plots provided in the right panels. All images were processed together so that the relative fluorescence intensity levels of the staining can be compared amongst different FLRT and Unc5 antibodies. Histogram images produced using ImageJ (*Schneider et al., 2012*). (**H**) Schematic summarizing expression pattern of each FLRT and Unc5 protein across IPL sublayers.

*Figure 6 continued on next page*

*Figure 6 continued*

The following figure supplements are available for figure 6:

**Figure supplement 1.** ELISA to test binding specificity of FLRT and Unc5 antibodies.

**Figure supplement 2.** Developmental analysis of FLRT and Unc5 expression in the IPL.

## Unc5C is a repulsive ligand for the FLRT2 receptor

Repulsive signaling of Unc5 in response to ligand binding has been well-established (for review of Netrin1-induced repulsion see *Moore et al., 2007*; for FLRT2-induced repulsion via Unc5D see *Yamagishi et al., 2011*). In our stripe assays we observe FLRT2-expressing retinal neurons that are repelled by Unc5C which is consistent with a model whereby Unc5C binding to FLRT2 induces repulsion in the FLRT2-expressing neuron; however, repulsive signaling downstream of FLRTs has not been reported. So we next asked whether Unc5C-FLRT2 interactions can induce repulsion in FLRT2-expressing retinal neurons by performing gain-of-function stripe assays. Using transient transfection, we ectopically expressed either full-length FLRT2-myc or full-length Unc5C-FLAG (control) in retinal neurons cultured on Unc5C stripes and monitored the response of neurons that expressed these exogenous proteins as assessed by anti-myc and anti-FLAG immunostaining, respectively. Importantly, this gain-of-function experiment was possible because only 11% of wild-type retinal neurons are repelled by Unc5C stripes (*Figure 5A*) and, as such, the vast majority of neurons are available to exhibit a gain-of-function phenotype.

We tested several commercially-available transfection reagents and found one that was capable of giving rise to ~10% transfection efficiency in our retinal neuron cultures (n=67/691 neurons transfected, see *Materials and methods*). We obtained 39 FLRT2-myc transfected neurons and observed that all 39 neurons were repelled by Unc5C stripes (n=39/328 neurons transfected; 15 coverslips) (*Figure 5L*). In our control transfections, we obtained 28 neurons that expressed Unc5C-FLAG and observed that 27/28 neurons grew permissively across the Unc5C stripes (n=28/363 neurons transfected; 13 coverslips) (*Figure 5M*). One neuron that ectopically expressed Unc5C-FLAG was repelled by Unc5C stripes. We hypothesize that this neuron is one of the 11% of wild-type neurons that is endogenously repelled by Unc5C. These data demonstrate that FLRT2 is sufficient to mediate repulsion in response to Unc5C and, as such, repulsive signaling can occur downstream of FLRT2.

## SACs express FLRT2 and are repelled by Unc5C

We next sought to identify which of the ~70 different subtypes of IPL-projecting neurons are the ones that express FLRT2 and are repelled by Unc5C. In retinal sections, FLRT2 expression co-localized with VAChT expression in S2/4 at P4 and P6 (*Figure 6B* and *Figure 6—figure supplement 2*). As such, we hypothesized that the FLRT2-expressing neurons are the same neurons that express VAChT – i.e. the starburst amacrine cells (SACs) which arborize in S2/4 between P0 and P3 (*Stacy and Wong, 2003*). To determine whether SACs express FLRT2 during and following arborization within S2/4, we performed *in situ* hybridization against *Flrt2* in sections at both P1 and P6 along with calbindin immunostaining which selectively stains SACs at these ages (*Kay et al., 2012*). Calbindin immunostaining was used to label SACs because VAChT immunoreactivity does not persist through the *in situ* hybridization protocol (nor does it label SAC cell bodies at P2-6). This analysis revealed that *Flrt2* is expressed by a subset of cells that includes: 1) SACs; 2) a sparse non-SAC population in the inner nuclear layer (INL) (presumably amacrines due to their laminar position close to the IPL and the fact that bipolar cells are not yet born at P1); and 3) a non-SAC population in the ganglion cell layer (GCL) that, based upon their large soma size, are likely to be retinal ganglion cells (*Figure 7A*). Notably, at P1, *Flrt2* expression is predominantly detected in ON SACs whose cell bodies reside in the GCL while, at P6, *Flrt2* expression is predominantly detected in OFF SACs whose cell bodies reside in the INL.

To confirm that FLRT2 protein is expressed in SACs, we performed FLRT2 immunostaining on cultured retinal neurons from a mouse strain that genetically expresses tdTomato specifically in SACs (*Chat-Cre::Rosa$^{LSL-tdTomato}$*) (*Sun et al., 2013*). It was necessary to use these transgenic mice to

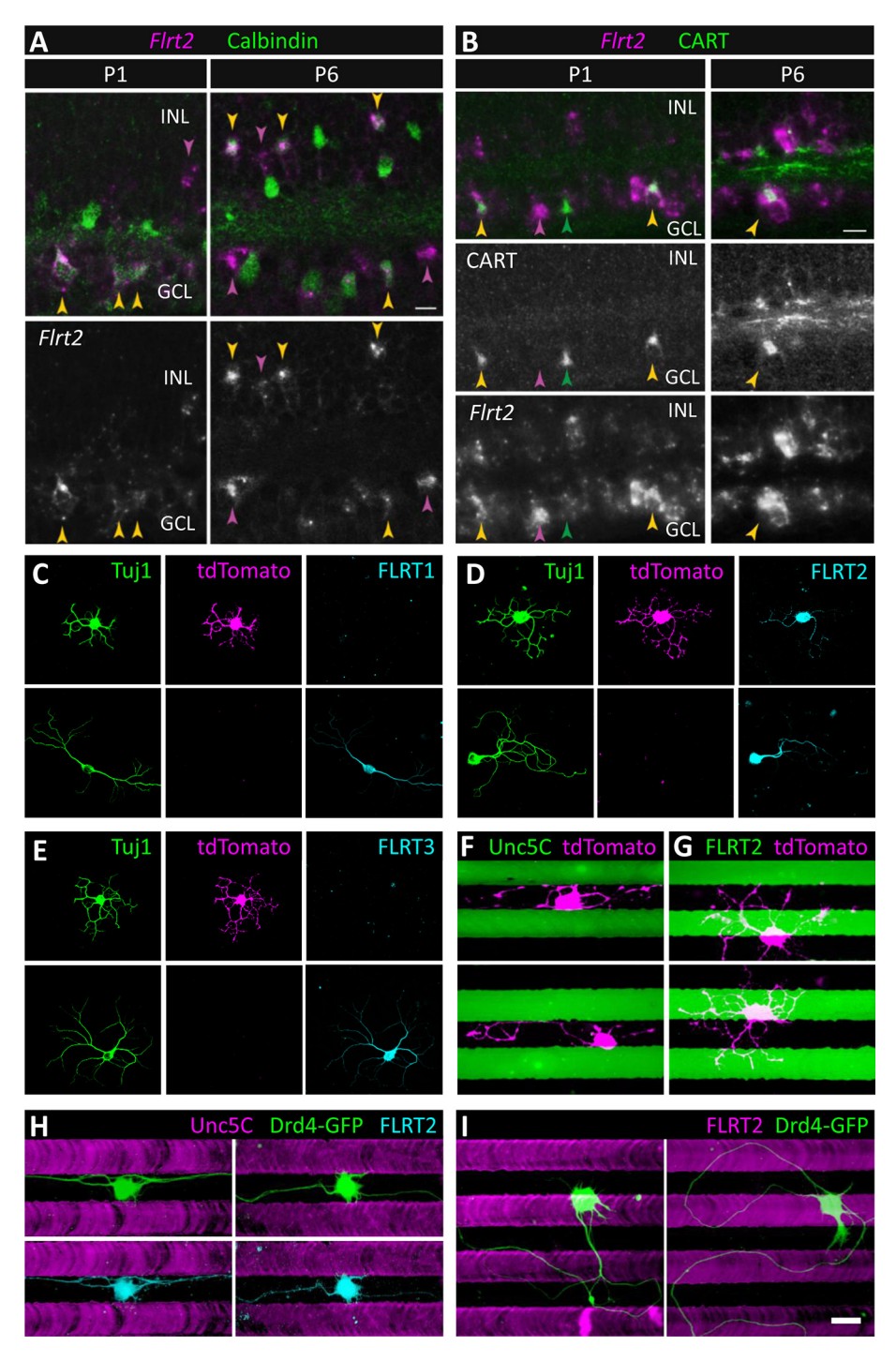

**Figure 7.** SACs and Drd4-GFP ooDSGCs express FLRT2 and are repelled by Unc5C. (**A**) *Flrt2* is expressed by SACs, a second amacrine population, and a subset of RGCs. *In situ* hybridization for *Flrt2* RNA (magenta) was combined with immunostaining for calbindin (green), a selective SAC marker at the ages shown (P1 and P6). Yellow arrows indicate *Flrt2*+ SACs. Cells in the inner nuclear layer (INL) expressing *Flrt2* but not calbindin (purple arrows) define a non-SAC *Flrt2*+ amacrine population. Non-SACs in the ganglion cell layer (GCL) are likely RGCs, based on their large soma size (purple arrows). Among SACs, *Flrt2* is detected predominantly in ON SACs (which reside in the GCL) at P1 whereas it is detected more readily in OFF SACs (which reside in the INL) at P6. However, ON SACs positive for *Flrt2* are observed at P6 (yellow arrow in GCL), suggesting that *Flrt2* is not selective for one SAC population over the other. (**B**) RGCs expressing *Flrt2* include direction-selective ganglion cells (DSGCs). Double staining for *Flrt2* and CART, an ooDSGC marker, at P1 and P6. Double-labeled cells (yellow arrows) are observed in the GCL. Not all ooDSGCs express *Flrt2*, however, as CART+ *Flrt2*– cells are also apparent (green arrows). Purple arrows indicate *Flrt2*+ cells that are not ooDSGCs; this group likely includes SACs. Scale bar, 10 μm.
*Figure 7 continued on next page*

*Figure 7 continued*

(**C**-**E**) SACs express FLRT2 protein. Dissociated SACs from P2 *Chat-Cre::Rosa^LSL-tdTomato^* mice that specifically express tdTomato (magenta) in SACs. Neurons were co-stained with an antibody against Tuj1 (green) and (**C**) FLRT1, (**D**) FLRT2, (**E**) FLRT3 (cyan). Only FLRT2 co-localized with tdTomato-positive SACs. SACs were also negative for Unc5s as shown in *Figure 7—figure supplement 1*. (**F**-**G**) tdTomato SACs (magenta) grown on Unc5C (**F**) or FLRT2 (**G**) stripes (green). Stripes were visualized by addition of PLL-FITC to the purified Unc5C or FLRT2 protein patterned. Unc5C (**F**) but not FLRT2 (**G**) repelled SACs. (**H**-**I**) Dissociated Drd4-GFP ooDSGCs (green) in culture harvested from P3 mice that specifically express GFP in ooDSGCs. (**H**) Drd4-GFP neurons on Unc5C stripes co-stained with an antibody against Tuj1 (green) and FLRT2 (cyan). (**I**) Drd4-GFP neurons on FLRT2 stripes stained with an antibody against Tuj1 (green). Neurons cultured 8 DIV. Scale bar, 30 µm.

The following figure supplement is available for figure 7:

**Figure supplement 1.** Expression of Unc5s in SACs.

visualize SACs in culture because the VAChT antibody that stains SACs in retinal sections does not stain cultured SACs (J.N.K., unpublished observations). Furthermore, it was necessary to perform FLRT2 immunostaining in dissociated cultured neurons because, in retinal sections, FLRT2 stains neurites in the IPL but not cell bodies in the adjacent INL and GCL (*Figure 6B* and *Figure 6—figure supplement 2*) thereby preventing identification of the cell(s) to which the FLRT2-positive neurites belong. Immunostaining of dissociated SACs harvested at P2 demonstrated that tdTomato-positive SACs express FLRT2 (n=47/47) but not FLRT1 (n=0/55) or FLRT3 (n=0/67) (*Figure 7C-E*). Consistent with our *in situ* hybridizations, we also observed non-SAC neurons that expressed FLRT2 (*Figure 7D*). As Unc5C expression localizes to S1/3/5 where SACs do not arborize (*Figure 6F* and *Figure 6—figure supplement 2*), we expected that SACs would not express Unc5C. Indeed, while a subset of tdTomato-negative neurons were immunoreactive for Unc5C, no Unc5C expression was observed in SACs (n=0/39) (*Figure 7—figure supplement 1*). Furthermore, none of the other Unc5s were expressed in SACs (*Figure 7—figure supplement 1*).

If the FLRT2-Unc5C interaction induces repulsion of SACs, we would expect FLRT2-expressing SACs to be repelled by Unc5C stripes in the *ex vivo* stripe assay. Indeed, we observed robust repulsion of SACs from Unc5C stripes (n=49/53, 92% repelled) (*Figure 7F*). In contrast, SAC processes crossed FLRT2 stripes indiscriminately (n=71/71, 0% repelled) (*Figure 7G*). Together these findings demonstrate that SACs express FLRT2 both during and after the developmental time when their neurites are becoming restricted to S2/4 and that SACs are repelled by Unc5C. Since SACs do not express FLRT1 and FLRT3, SAC repulsion by Unc5C could be due to interactions with FLRT2. These data suggest that repulsive FLRT2-Unc5C interactions may contribute to laminar organization of SAC neurons in the developing IPL.

## ON-OFF direction-selective ganglion cells express FLRT2 and are repelled by Unc5C

By *in situ* hybridization we found that *Flrt2* is expressed in a non-SAC population in the GCL (*Figure 7A*). Direction-selective ganglion cells (DSGCs) arborize dendrites in S2/4 and are the post-synaptic partners of SACs (*Demb, 2007*; *Wei and Feller, 2011*; *Vaney et al., 2012*; *Masland, 2012*). We therefore wondered whether DSGCs might also express *Flrt2*. To test this idea, we combined *Flrt2 in situ* hybridization with immunostaining against the neuropeptide CART (cocaine- and amphetamine-regulated transcript), which stains the most numerous category of DSGCs, ON-OFF DSGCs (ooDSGCs) (*Kay et al., 2011a*). CART is a selective (though not exclusive) marker for ooDSGCs (*Kay et al., 2011b*; *Ivanova et al., 2013*). We observed that about half of CART-immunoreactive cells are *Flrt2*-positive (n=12/23 CART+*Flrt2*+, n=11/23 CART+*Flrt2*–) suggesting that a subset of ooDSGCs expresses FLRT2 (*Figure 7B*).

As ooDSGCs exhibit S2/4 laminar restriction, we next asked whether ooDSGCs, like SACs, express FLRT2 protein and are repelled by Unc5C stripes. To test this we cultured neurons from a mouse strain that genetically expresses GFP under control of the dopamine receptor 4 promoter (*Drd4-GFP*) in a subtype of ooDSGCs that prefer posterior motion (*Gong et al., 2003*; *Huberman et al., 2009*; *Kay et al., 2011a*). The Drd4-GFP cells were encountered in our cultures only rarely, perhaps because our cultures were not optimized for RGC survival, or because they are a remarkably sparse cell type comprising ≤5% of ganglion cells which are themselves only 1% of

retinal neurons (*Kay et al., 2011a*). Nevertheless, when healthy Drd4-GFP neurons were identified, we observed that they expressed FLRT2 and were repelled by Unc5C stripes (n=7/7, 100% repelled; 7 coverslips) (*Figure 7H*) but not by FLRT2 stripes (n=10/10, 0% repelled; 7 coverslips) (*Figure 7I*). These data suggest that at least one subtype of DSGCs may utilize repulsive FLRT2-Unc5C interactions to achieve laminar restriction in the developing IPL.

## Discussion

The IPL is innervated by ~70 different subtypes of neurons that organize into a distinct, stereotyped laminar structure. The level of molecular recognition required at the cell surface to achieve this complex circuitry is likely to be staggering. To begin to understand how this molecular choreography is achieved on a global level, we need to be able to consider the complete IPL extracellular interactome in the context of cell subtype-specific expression and functional growth responses. Our approach is based on the widely-accepted notion that neuronal subtype-specific differences in composition and/or levels of cell surface and secreted proteins underlie the ability of neurons to recognize and respond to one another and the environment in a highly precise fashion. As such, it is the differentially-expressed proteins, the unique cell surface identity of each neuronal subtype, that reside at the heart of recognition specificity.

Here we present the first extracellular receptor-ligand screen comprising candidate cell surface and secreted proteins selected due to differential expression among multiple cell subtypes as assessed by gene profiling. Using this directed approach, we identified high confidence candidates for mediating cell recognition events in the developing IPL and then conducted a candidate-based biochemical screen. We identified new receptor-ligand pairs and, as such, have begun to characterize the extracellular interactome in the developing retina.

### Identification of FLRT and Unc5 protein families as candidate IPL lamination molecules

The results of our binding screen drew our attention to FLRTs and Unc5s. We discovered that members of these protein families are expressed in strikingly specific laminar patterns during early IPL development. Using stripe assays, we found that all members of these families except Unc5A and Unc5B are capable of eliciting attractive and/or repulsive behavior from subsets of retinal neurons. Notably, Unc5A and Unc5B also showed the least laminar specificity in their IPL expression patterns. These two features of Unc5A and Unc5B biology suggest that they are unlikely to play a role in IPL lamination. By contrast, the other members of these two families are excellent candidates to mediate IPL lamination based on their expression patterns, bioactivities and receptor-ligand interactions that we report here.

The expression patterns of FLRT2 and Unc5C are remarkably complementary in the developing IPL, suggestive of a repulsive role for this receptor-ligand pair. Consistent with this notion, we found that neurons expressing FLRT2 are repelled by Unc5C and, conversely, neurons expressing Unc5C are repelled by FLRT2. Using transfected primary neurons, we demonstrated that ectopic expression of FLRT2 is sufficient to mediate repulsion in response to Unc5C. While we cannot rule out the possibility that this response to Unc5C arises due to the presence of another cell surface protein(s) that gets recruited in *cis* by exogenous FLRT2 expression, taken together our data suggest that FLRT2-Unc5C interactions can induce repulsion in a subset of primary retinal neurons.

### Interactions between Semaphorin, Plexin and Neuropilin proteins

Sema, Plxn and Nrp proteins comprise large numbers of diverse cell recognition proteins involved in neural circuit formation and an ever-increasing list of cell biological processes (for review see *Yoshida, 2012*; *Gu and Giraudo, 2013*). While many binding partners within these families have been described, a comprehensive study of all Sema-Nrp and Sema-Plxn pairs has never been conducted. We included the complete families because our microarray data demonstrated that many members are differentially expressed in different subtypes of IPL neurons. Additionally, at the time we were selecting candidates for our screen, Kolodkin and colleagues reported that Sema5A and Sema5B interactions with PlxnA1 and PlxnA3 play a role in laminar organization in the developing mouse IPL (*Matsuoka et al., 2011*). As such, we hypothesized that other family members are involved and reasoned that understanding the complete interaction network is necessary for evaluating genetic

phenotypes *in vivo*. Previously, Sema3s were believed to require Nrp1 for signaling through PlxnA co-receptors (*Tamagnone et al., 1999*) and Cntn2 was believed to interact with Sema3A only indirectly through *cis* interactions between Cntn2 and Nrp1 (*Dang et al., 2012*). Direct protein-protein interactions observed in our screen between Sema3A-Cntn2 and Sema3A-PlxnA4 suggest that Sema3A may be able to signal directly through these receptors in the absence of Nrp1 (*Figure 3B*). The additional interaction partners we identified will thus enable the field to better understand how the interplay among Semas-Plxns-Nrps, as well as other potential Sema receptors such as Cntn2 and PlxnA4, contribute to laminar organization of the IPL and other cellular responses in a variety of different systems.

## All FLRT and Unc5 family members interact heterophilically with one another

The three FLRTs and four Unc5s represent 12 potential heterophilic receptor-ligand pairs. Prior to our screen, four pairs had been reported amongst varying combinations of *Xenopus* and mouse proteins (FLRT1-Unc5B, FLRT2-Unc5D, FLRT3-Unc5B and FLRT3-Unc5D) (*Karaulanov et al., 2009*; *Sollner and Wright, 2009*; *Yamagishi et al., 2011*). Using a variety of binding assays, we observed interactions between all FLRTs and all Unc5s. Further confirmation that the eight additional FLRT-Unc5 pairs we observed are biologically-relevant has been provided by Seiradake et al. who recently reported several of these interactions (*Seiradake et al., 2014*).

FLRTs and Unc5s are broadly expressed in the developing nervous system as well as in other tissues. While in some regions FLRTs and Unc5s exhibit striking cell-type-specific expression patterns (including the cortex, hippocampus and the developing retina as we have shown here), in other areas multiple FLRTs and Unc5s are expressed in overlapping regions (*Haines et al., 2006*; *Gong et al., 2009*; *Yang et al., 2013*; *Seiradake et al., 2014*). As such, the promiscuous binding of all FLRTs to all Unc5s seemingly presents a conundrum. Based upon the observed binding properties, a FLRT2-expressing neuron might well interact with all neurons that express any one of the four Unc5s. As such, how can FLRT-Unc5 interactions provide recognition specificity? Does promiscuous binding reduce the total possible number of distinct FLRT-Unc5 binding specificities from 12 (i.e. 3 FLRTs x 4 Unc5s) to one (i.e. FLRT-Unc5)? Our experiments (*Figure 2C*) and those of others (*Seiradake et al., 2014*) have demonstrated that different FLRT-Unc5 pairs exhibit differences in binding affinity (while our binding curves plateau due to saturated levels of detection and therefore preclude the determination of binding constants, the qualitative determination that there are differences can be inferred from the shifting of curves relative to one another along the x-axis). We speculate that these differences in binding affinity contribute to recognition specificity. The diverse cadherin family of homophilic and heterophilic cell surface proteins provides a classic example where this is the case. As with FLRTs and Unc5s, several members of the cadherin family exhibit similar levels of promiscuous homophilic and heterophilic binding in cultured cell-based assays but, when binding constants are determined using SPR or analytical ultra centrifugation, differences in binding affinity are observed which, in turn, mediate the sorting of cells into different tissues *in vivo* (*Katsamba et al., 2009*).

## FLRT-FLRT interactions likely occur in *cis*

Conflicting reports have been published regarding whether or not FLRTs engage in homophilic interactions (*Karaulanov et al., 2006*; *Yamagishi et al., 2011*; *Seiradake et al., 2014*; *Lu et al., 2015*). Similar to previous experiments that failed to detect binding of soluble FLRT ectodomains to FLRT-expressing cells in culture (*Yamagishi et al., 2011*) or FLRT-mediated cell aggregation (*Lu et al., 2015*), we did not observe FLRT homophilic interactions in our biochemical screen or cell aggregation assay. Furthermore, in our stripe assays, FLRT2-expressing SACs and Drd4-GFP ooDSGCs did not respond to FLRT2 stripes. A recent study reported that FLRT homophilic binding is difficult to detect *in vitro* due to very low binding affinity and is highly sensitive to experimental conditions (*Seiradake et al., 2014*). When measured using surface plasmon resonance, homophilic binding of FLRTs was below the sensitivity of detection (~100 µM) and, in SEC-MALS experiments, a minor increase in molecular weight (from ~70 kDa to ~80 kDa) was seen with increasing concentration, but no well-defined FLRT dimer fraction was observed. In addition, detection of FLRT-mediated homophilic cell aggregation required five days of continuous cell shaking, a time period considerably longer than standard protocols which typically monitor cell aggregation after shaking for 1−4 hours.

In crystal structures of a portion of the FLRT2 and FLRT3 extracellular domains, conserved lattice contacts were observed between *cis*-oriented FLRT proteins (*Seiradake et al., 2014*). Mutations at this interface impaired tangential spread of pyramidal neurons between adjacent cortical columns *in vivo* which the authors interpreted as a resulting from a defect in attractive FLRT homophilic binding. Subsequent structural and biochemical studies by Lu et al. investigating interactions between FLRT and latrophilin, a cell surface adhesion-type G-protein-coupled receptor, demonstrated that, while the FLRT mutant exhibits a decrease in dimerization via size-exclusion gel filtration, binding of the FLRT mutant to latrophilin is completely abolished (*Lu et al., 2015*). These findings, in addition to the authors' inability to detect FLRT homophilic binding between cells, led them to conclude that the FLRT homodimer likely occurs in *cis* and that the *in vivo* pyramidal neuron phenotype may be due to a defect in FLRT-latrophilin binding. In our stripe assays we observe subpopulations of primary retinal neurons that are attracted to FLRT1 and FLRT3 stripes. As latrophilins are expressed in the retina (*Arcos-Burgos et al., 2010*) (J.N.K., unpublished observations), it will be interesting to determine whether attraction of these neurons is mediated by FLRT interactions with neuronally-expressed latrohpilin or another yet-unidentified *trans* interaction partner.

## Repulsive signaling may be a conserved function of all Unc5 receptors

Repulsive signaling induced by FLRT2 ligand binding to Unc5D-expressing pyramidal neurons modulates radial migration in the developing mouse cortex (*Yamagishi et al., 2011*). Furthermore, FLRT3 induces repulsion of Unc5B-expressing intermediate thalamic explants *ex vivo* (*Seiradake et al., 2014*). In both of these cases, neurons expressing Unc5s are repelled by FLRT ligand demonstrating that signaling downstream of Unc5 induces repulsion in the Unc5-expressing cell. Consistent with these findings, we observed that Unc5C-expressing retinal neurons are repelled by FLRT2. These data suggest that, in addition to Unc5B and Unc5D, signaling downstream of Unc5C can elicit a repulsive response.

## FLRT2-Unc5C interactions may induce bidirectional repulsive signaling

We observed that FLRT2-expressing SACs and Drd4-GFP ooDSGCs are repelled by Unc5C ligand. These observations are consistent with a mechanism whereby binding of Unc5C ligand to FLRT2 receptor induces repulsive signaling in the FLRT2-expressing cell. Using a gain-of-function stripe assay, we found that FLRT2 expression is sufficient to elicit a repulsive response to Unc5C ligand. These findings suggest the intriguing possibility that a bidirectional mechanism of repulsive signaling can occur whereby FLRT2-Unc5C interactions induce repulsion in both FLRT2- and Unc5C-expressing cells. A mechanism of bidirectional signaling has been well characterized between Eph receptors and their ephrin ligands (for review see *Park and Lee, 2015*). Such a mechanism of Unc5C-FLRT2 mutual repulsion would provide an elegant and efficient molecular solution for directing laminar organization/restriction of both FLRT2- and Unc5C-expressing neurons into adjacent layers, S2/4 and S1/3/5, respectively, during development of the IPL. Our future studies will be aimed at identifying and characterizing the neuronal subtype(s) that arborizes in S1/3/5 and expresses Unc5C to determine whether they are repelled by FLRT2 and if they are necessary to ensure laminar restriction of SACs and ooDSGCs in S2/4. Furthermore, as additional subtypes that we have not yet characterized also express FLRT2, other neurons have the potential to utilize FLRT2 for laminar organization either through interactions with Unc5C or other FLRT2 binding partners.

## FLRT2 and the development of retinal direction-selective circuitry

IPL sublayers contain axons and dendrites of retinal neurons devoted to specific visual processing tasks (*Masland, 2012*). By projecting to the same sublayer, circuit partners interact specifically with each other, facilitating appropriate synaptic partner choices. A striking example is the retinal circuit that detects image motion, the so-called direction-selective (DS) circuit, which comprises cofasciculated arbors of SACs and ooDSGCs stratified in IPL sublayers S2 and S4. Precise inhibitory connections from SACs onto DSGCs regulate DSGC firing in response to motion in particular directions, producing direction-selective responses (*Demb, 2007*; *Wei and Feller, 2011*; *Vaney et al., 2012*; *Masland, 2012*). The mechanisms mediating the initial assembly of these IPL sublayers, or the co-recruitment of SAC and ooDSGC to those layers, are not known. The laminar choices of ON and OFF SACs are influenced by repulsive interactions between Plxn2 and Sema6A (*Sun et al., 2013*).

However, in PlxnA2$^{-/-}$ and Sema6A$^{-/-}$ mutants, most SAC dendrites still assemble in the correct sublamina and even when SACs make errors they still target to S2 or S4 (*Sun et al., 2013*). This suggests that an additional molecular mechanism(s) functions in parallel to mediate precise laminar restriction of SACs. Here we show that SACs and at least one subset of ooDSGCs (the Drd4-GFP population) express FLRT2 and are repelled by Unc5C. We propose that these (and perhaps other) direction-selective circuit neurons become laminar-restricted in S2/4, and/or maintain their laminar restriction once formed, due to repulsive interactions with Unc5C expressed on neighboring neurites in S1/3/5. Definitive evidence that SACs and/or Drd4-GFP cells require FLRT2 and Unc5C for laminar targeting in S2/4 awaits genetic loss-of-function analyses. Nevertheless, our results suggest that evolution may have co-opted the same repulsive mechanism in both pre- and post-synaptic cells as a strategy for ensuring they both arborize in close spatial proximity to one another, thereby facilitating interactions between synaptic partners and limiting opportunities for inappropriate connections with neurons devoted to different visual processing tasks.

## Conclusions

Here we present an integrated systems-level approach using cell subtype-specific gene profiling to drive candidate-based, high-throughput, biochemical receptor-ligand screening. Using this approach, we demonstrate that, in addition to genetic screens, biochemical screens provide another strategy for identifying recognition proteins that play a role in facilitating the laminar organization that underlies visual function. However, this study represents merely the tip of the iceberg. Our biochemical screen sampled only a small fraction of the recognition proteins present in a limited number of neuronal subtypes in the developing IPL. Here we present data that support a model for how a single receptor-ligand interaction contributes to the laminar organization of two subtypes of neurons. However, our ultimate goal is to understand lamination on a global scale. We are optimistic that combining 1) inclusive gene profiling data gathered from each of the ~70 different IPL neuronal subtypes (for which numerous more markers are now available) with 2) larger-scale biochemical screens aimed at identifying the entire IPL extracellular interactome, we can elaborate a comprehensive view of how laminar organization develops in the mouse IPL.

## Materials and methods

### Bioinformatics and microarray analysis

Microarrays for 13 different subtypes of IPL neurons were performed as described (*Kay et al., 2011b*; *Kay et al., 2012*) (NCBI Gene Expression Omnibus; accession GSE35077). A variety of on-line tools and databases were used to identify differentially-expressed genes that encode transmembrane, GPI-linked and secreted proteins. The details of these methods are described in *Figure 1—figure supplement 1*.

### Antibodies

Antibodies used in this study include: mouse anti-PLAP (Thermo Fisher Scientific; Waltham, MA), mouse anti-human IgG1-Fc-HRP (Serotec; Raleigh, NC), mouse anti-myc (Abcam; UK, 1:1000), mouse anti-FLAG (Abcam, 1:1000), chicken anti-GFP (Abcam, 1:6000), goat anti-FLRT1 (R&D Systems; Minneapolis, MN, 1:25), rabbit anti-FLRT2 (Abcam, 1:25), goat anti-FLRT3 (R&D Systems, 1:50), goat anti-Unc5A (R&D Systems, 1:25), rabbit anti-Unc5B (Santa Cruz Biotechnology; Santa Cruz, CA, 1:200), rabbit anti-Unc5C (Santa Cruz Biotechnology, 1:50), goat anti-Unc5D (R&D Systems, 1:100), mouse anti-His-HRP (Qiagen; Germany, 1:5000), goat anti-Human IgG (H+L) DyLight 680 (Rockland; Limerick, PA, 1:4000), guinea pig anti-vesicular acetylcholine transporter (VAChT) (EMD Millipore; Hayward, CA, 1:500), mouse anti-neuronal class III beta-tubulin (Tuj1) (Covance; Princeton, NJ, 1:1000), rabbit anti-cocaine- and amphetamine-regulated transcript (CART) (Phoenix Pharmaceuticals; Burlingame, CA, 1:2000), rabbit anti-calbindin (Swant Inc; Switzerland, 1:5000).

### Cell lines

HEK293T and CHO.K1 cells were cultured according to ATCC guidelines.

## Animals

C57Bl/6 mice (Harlan) were used for wild-type retinal section immunostaining and primary retinal neuron cultures. *Chat-Cre::Rosa*[LSL-tdTomato] mice were generated by crossing a tdTomato driver line (B6.129S6-*Chat*[tm1(cre)lowl]/J × B6.129S6-*Gt(Rosa)26Sor*[tm9(CAG-tdTomato)Hze]/J, Jackson Labs; Bar Harbor, ME) with a mouse that has an IRES-Cre recombinase downstream of the endogenous choline acetyl transferase gene (*Ivanova et al., 2010*). *Chat-Cre::Rosa*[LSL-tdTomato] mice express fluorescent protein in SACs. Dopamine receptor D4-GFP (Tg(*Drd4-GFP*)W18Gsat) mice were obtained from Mutant Mouse Regional Resource Center-University of North Carolina (https://www.mmrrc.org/catalog/sds.php?mmrrc_id=231) (*Gong et al., 2003*). Genotypes were identified using genomic PCR. All animal procedures were approved by the University of California, Berkeley (Office of Laboratory Animal Care (OLAC) protocol #R308) and they conformed to the National Institutes of Health *Guide for the Care and Use of Laboratory Animals*, the Public Health Service Policy and the Society for Neuroscience Policy on the Use of Animals in Neuroscience Research.

## Cloning

Retinal genes were PCR amplified from mouse retinal cDNA. Upstream and downstream primers contained NotI and SpeI or AscI sites (*Figure 1—source data 1*), respectively, which were used to subclone into two pCMVi vectors (gift of John Ngai), pCMVi-*[extracellular region]*-AP-6X-His and pCMVi-*[extracellular region]*-Fc-6X-His. Mouse *Dscam, Dscaml1, Sdk1* and *Cntn* genes were subcloned from existing plasmids (*Yamagata and Sanes, 2008*). Full-length versions of FLRT1-3 and Unc5A-D were cloned from retinal cDNA into a derivative of the pTT3 vector (*Bushell et al., 2008*) and into pUB using downstream primers that introduce C-terminal myc and FLAG epitope tags, respectively. All plasmids used in this study have been submitted to Addgene (Cabridge, MA).

## Recombinant protein production

Fc-6X-His- and AP-6X-His-tagged recombinant proteins were expressed by transient transfection of HEK293T cells grown in media containing 10% Ultra-Low IgG fetal bovine serum (Invitrogen; Carlsbad, CA) using linear polyethylenimine (PEI) transfection reagent (Thermo Fisher Scientific). For 15 cm plates, 32 μg of plasmid DNA and linear PEI ($C_f$=40 μg/ml) was added to 3.2 ml Opti-MEM (Invitrogen), vortexted briefly, incubated for exactly 10 minutes at room temperature and added dropwise onto cells. Culture media was harvested 6 days post transfection. The amount of Fc- and AP-tagged proteins in the media was quantified as described previously (*Wojtowicz et al., 2007*). For stripe assays, 6X-His-tagged proteins were purified using TALON metal affinity resin (Clontech Laboratories; Mountain View, CA) and quantified using the Bradford assay as described previously (*Wojtowicz et al., 2004*).

## Binding screen

AP and Fc tags were specifically chosen for their ability to homodimerize. This forces the attached extracellular domain to adopt a dimer conformation. Further clustering of the dimerized proteins is achieved using monoclonal anti-AP and anti-Fc antibodies at limiting concentrations, thereby forcing saturation of the antibodies with a dimer bound to each of the antibody's two binding sites – thus inducing a tetrameric conformation. The technical aspects of the binding screen were modified from *Wojtowicz et al., 2007* as follows: AP-tagged protein was used at 33 U/ul (where a unit [U] is equivalent to the activity of 10 pg of purified calf intestinal phosphatase (Thermo Fisher Scientific Pierce)) and Fc-tagged protein was used at 140 nM. This was necessary to convert the assay from one that tested *Drosophila* proteins expressed in *Drosophila* S2 cells to one that tests mammalian proteins produced in HEK293T cells. Background ($Abs_{650nm}$ = 0.064) was determined using wells containing all binding reaction components with mock culture media in place of AP-tagged culture media. Background-subtracted data were deposited to the Dryad database *Visser et al., 2015*.

## Cell aggregation assay

CHO.K1 cells were co-transfected with pTT3-FLRT-myc + pGreen or pTT3-Unc5-FLAG + dsRed plasmids at a 5:1 ratio using TransIT-CHO transfection reagent (Mirus Bio; Madison, WI) according to the manufacturer's protocol. Cells were incubated at 37°C and 5% $CO_2$ overnight, harvested with trypsin for exactly 5 minutes, resuspended in aggregation media (CHO.K1 media containing 70 U/ml

DNAse I and 2 mM EGTA) and counted. FLRT-myc/GFP and Unc5-FLAG/RFP cells ($0.5 \times 10^5$ each in 250 ul) were mixed together in a 24-well ultra-low adhesion plate (Corning Inc; Corning, NY) and incubated for four hours in a 37°C, 5% $CO_2$ incubator on a belly dancer mixer at 90 rpm. Cells were diluted 1:5 in aggregation media and 100 ul was added to two 35 mm glass-bottom dishes (MatTek Corp; Ashland, MA). Clusters containing >10 cells were counted using an Axiovert S100 fluorescence microscope (Carl Zeiss; Germany).

## Microfluidic device fabrication

Microfluidic devices were designed using the AutoCAD program (AutoDesk; San Rafael, CA). The design included nine groupings of ten channels. Channels were 30 µm wide, 100 µm high and separated from one another by 30 µm. Each grouping was separated by 150 µm. Microfluidic device features were fabricated using SU8 photoresist on a silicon wafer (Stanford Foundry; Stanford University, Palo Alto, CA) and coated with Teflon for quick feature release. Features were then transferred into polyurethane casting masters (Smoothcast 326). Devices were produced as follows: Poly-dimethyl-siloxane (PDMS, SYLGARD) was mixed in a 10:1 base to crosslinker ratio, poured into casting masters, degassed overnight and let cure at 37°C for a minimum of 24 hours. After release peel from the casting master, 1.2 mm inlet and outlet holes were punched (Ted Pella Inc; Redding, CA) and devices were mounted feature side up on glass slides before wrapping in aluminum foil and autoclaving for 10 minutes. Following autoclaving, devices were allowed to dry overnight at room temperature.

## Stripe assay

Glass coverslips (12 mm Assistant-Brand, Carolina; Burlington, NC) were washed with 70% ethanol for 7 days with ethanol changed every day and then stored in 70% ethanol. Upon removal from ethanol, coverslips were rinsed thoroughly with water, coated sequentially with 25 µg/ml poly-D-lysine (Sigma-Aldrich; St Louis, MO) and 50 µg/ml laminin (Sigma-Aldrich). Microfluidic devices were applied to coverslips and desiccated to strengthen seal. Stripes were prepared by pulling protein solutions through microfluidic devices using a vacuum at 7 psi. Protein solutions contained 100 µg/ml purified protein (FLRT-Fc-6X-His, Unc5-Fc-6X-His or laminin), mixed with 100 µg/ml BSA-TRITC or PLL-FITC (to visualize the stripes). Protein solutions were incubated in devices at 37°C in a humidified chamber overnight and then wet-peeled in autoclaved milliQ water and stored in 1X PBS until use.

Dissociated retinal neurons were prepared using a modified version of a protocol developed by Ben Barres (*Barres et al., 1988*). Retinas from P6 (wild type; three independent experiments), P2 (*Chat-Cre::Rosa*^LSL-tdTomato^; three independent experiments) and P3 (*Drd4-GFP*; two independent experiments) mice were quickly dissected from the eyecup into cold D-PBS (GE Healthcare HyClone; Logan, UT), followed by digestion in D-PBS containing (per 500 ml) 165 units of papain (Worthington Biochemical; Lakewood, NJ), 2 mg of N-Acetyl-L-Cysteine (Sigma-Aldrich), 8 µl 1N Sodium Hydroxide (Sigma-Aldrich) and 0.4 mg DNase (Worthington Biochemical) for 45 minutes at 37°C. The retinas were gently triturated in low-ovomucoid (Worthington Biochemical) then high-ovomucoid (Worthington Biochemical), each trituration step followed by a 10 minute spin at 1000 rpm. Cells were resuspended in panning buffer (0.02% BSA in D-PBS, 5 µg/ml insulin), passed through a 40 µm cell strainer and then incubated for 30 minutes in a 15 cm petri dish coated with lectin I from Bandeiraea simplicifolia (BSL-1) (Vector Laboratories; Burliname, CA) to deplete macrophages (with vigorous shaking at 15 and 30 minutes to remove non-specifically attached cells). The supernatant was harvested, passed through a 40 µm cell strainer and $0.5 \times 10^5$ cells were seeded ($1 \times 10^5$ for Drd4-GFP) per well of 24-well plates onto glass coverslips containing purified protein stripes. Cells were seeded into 750 µl neurobasal-based culture medium (Invitrogen) containing 50 U/ml penicillin, 50 µg/ml streptomycin (Invitrogen), 5 µg/ml insulin (Sigma-Aldrich), 1 mM sodium pyruvate (Invitrogen), 100 µg/ml transferrin (Sigma-Aldrich), 100 µg/ml crystalline BSA (Sigma-Aldrich), 60 ng/ml progesterone (Sigma-Aldrich), 16 µg/ml putrescine (Sigma-Aldrich), 40 ng/ml sodium selenite (Sigma-Aldrich), 160 µg/ml triiodo-thyronine (Sigma-Aldrich), 2 mM L-glutamine (Sigma-Aldrich), B-27 Supplement (Invitrogen), 50 µg/ml N-Acetyl Cysteine (Sigma-Aldrich), 50 ng/ml brain derived neurotrophic factor (BDNF) (Peprotech; Rocky Hill, NJ), 10 ng/ml ciliary neurotrophic factor (CNTF) (Peprotech) and 10 nM forskolin (Sigma-Aldrich). Cultures were incubated at 37°C, 5% $CO_2$.

Every 2–3 days, half of the volume of the media in each well was removed and replaced with fresh media. Neurons were allowed to grow for 4–8 days.

For gain-of-function stripe assays, neurons were transfected in three independent experiments approximately 24 hours post seeding as follows using Attractene transfection reagent (Qiagen). 0.2 µg of plasmid DNA and 0.5 µl of Attractene was added to Opti-MEM in a final volume of 60 µl, incubated 15 minutes at room temperature and added dropwise onto cells. Following transfection, cells were allowed to grow as described above. Note that for expression in primary retinal neurons, the FLRT2-myc and Unc5C-FLAG transgenes were moved from pTT3 (vector used for cell aggregation assays) into the pUB vector, bearing the human Ubiquitin-C promoter. For reasons that are unclear to us, transfection of the pGreen vector gave rise to an ~10% transfection efficiency as determined by the number of Tuj1+/GFP+ vs Tuj1+/GFP- neurons but transfection with pTT3-FLRT2-myc yielded hardly any FLRT2-myc+ cells. When we moved the FLRT2-myc transgene into pUB, we obtained robust FLRT2-myc expression in ~10% of neurons. As such, expression vector choice can have a significant effect on transfection results and, in this case, was crucial for the success of the experiment.

## Immunohistochemistry

Retinas were dissected from P2, P4 and P6 wild-type mice, fixed 1.5 hours (P2 and P4) or 45 minutes (P6) in 4% paraformaldehyde at 4°C, equilibrated in 30% sucrose until retinas sank (2-3 hours), immediately embedded in O.C.T. (Tissue-Tek), frozen on dry ice and sectioned immediately or stored at −80°C until sectioning. Cryostat sectioning (10 µm) was performed using a Microm HM550 (Thermo Fisher Scientific). Sections were blocked 1 hour in 1X PBS containing 2% normal donkey serum, 2% BSA, 4% Triton X-100, 0.4% SDS (blocking buffer) and incubated with primary antibodies in blocking buffer overnight at 4°C. Secondary antibodies were incubated in blocking buffer for 45 minutes at room temperature. Sections were imaged using a Nikon Eclipse E600 fluorescence microscope (Nikon; Japan). Primary neurons and CHO.K1 cells were fixed in ice cold 4% paraformaldehyde/1X PBS for 15 minutes, blocked 30 minutes and incubated with primary antibodies overnight at 4°C (blocking buffer for CHO.K1 cells was 1X PBS containing 2% normal donkey serum, 2% BSA, 0.05% Triton X-100). Secondary antibodies were incubated 2 hours at room temperature. Primary neurons were imaged using a Nikon Eclipse E600 fluorescence microscope with the exception of triple-labeling experiments (i.e. when far red secondary antibodies were used) and then neurons were imaged using a Zeiss LSM 710 AxioObserver confocal microscope. CHO.K1 cells were imaged using a Zeiss Axiovert S100 fluorescence microscope.

## Double staining by *in situ* hybridization and immunohistochemistry

Full-length *Flrt2* cDNA (NCBI accession #BC096471) was obtained from GE Dharmicon (Lafayette, CO) in vector pCMV-Sport6. Sequencing confirmed presence of the correct insert. Plasmid was linearized at the 5' end of the insert and antisense digoxigenin-labeled RNA probes (DIG RNA labeling mix) (Roche Diagnostics; Switzerland) were synthesized using a T7 site present in the vector (MAXIscript kit, Thermo Fisher Scientific). The probes were purified on a G50 spin column (GE Healthcare) and hydrolysed at 60°C in bicarbonate buffer (40 mM $NaHCO_3$, 60 mM $Na_2CO_3$) to an expected size of 500 bp. P1 and P6 retinas were quickly dissected from the eyecup in ice-cold Hank's balanced salt solution buffered by 10 mM HEPES, fixed in 4% paraformaldehyde/1X PBS for 90 minutes on ice, washed twice with 1X PBS, and sunk in 30% sucrose/1X PBS for 1 hour. Immediately upon sinking, tissues were frozen in TFM (Triangle Biomedical Sciences; Durham, NC) and stored at −80°C until sectioning at 20 µm on a cryostat. *In situ* hybridization was performed on retinal sections as described (*Kay et al., 2011b*; *Yamagata et al., 2002*). Probes were detected with peroxidase-coupled anti-digoxigenin followed by a Cy3-tyramide color reaction. After the color reaction, slides were washed at least 4 times over 2 hours in 1X PBS. They were then subjected to antibody labeling as follows. Slides were incubated in blocking solution (1X PBS containing 3% donkey serum and 0.3% Triton X-100) for 30 minutes at room temperature. Primary antibodies, diluted in blocking solution, were applied overnight at 4°C. Slides were washed twice in 1X PBS and stained with donkey anti-rabbit secondary antibodies conjugated to Alexa-488 (Jackson Immunoresearch; West Grove, PA, 1:1000). Retinas from four different mice were used at each age, and were stained in two independent experiments.

## Acknowledgements

We thank Thomas Holton (University of California, Los Angeles) for bioinformatics assistance, Matthew Nicotra and Uma Karadge (University of Pittsburgh) for help with cell aggregation assays, Ben Barres (Stanford University) and Marla Feller (University of California, Berkeley) and many members of their laboratories for much time and assistance with culturing retinal neurons, Marla Feller and Xiaokang (Paley) Han for mouse husbandry, John Ngai and members of his laboratory for histology assistance, helpful discussions and use of their Nikon Eclipse E600 fluorescence microscope. We acknowledge Aarushi Kalaimani for help with cloning. Several UC Berkeley core facilities were utilized including the High-Throughput Screening Facility, CRL Molecular Imaging Facility (supported by the Gordon and Betty Moore Foundation), Cell Culture Facility and Biomolecular Nanotechnology Center. Stanford Foundry fabricated SU-8 wafer masks. We thank Daisuke Hattori, John Ngai, Marla Feller, Joshua Sanes and Larry Zipursky for critical feedback and suggestions on an early version of the manuscript. WMW, JJV, YC, SCP, ABC, BP and SSM were supported by a generous fellowship provided by the Bowes Foundation. JNK was supported by NEI (1R01EY024694), Karl Kirchgessner, E. Matilda Ziegler, Pew, McKnight and Sloan Foundations.

## Additional information

### Funding

| Funder | Grant reference number | Author |
|---|---|---|
| Bowes Research Foundation | | Jasper J Visser<br>Yolanda Cheng<br>Steven C Perry<br>Andrew Benjamin Chastain<br>Bayan Parsa<br>Shatha S Masri<br>Woj M Wojtowicz |
| National Eye Institute | 1R01EY024694 | Jeremy N Kay |
| Karl Kirchgessner Foundation | | Jeremy N Kay |
| E. Matilda Ziegler | | Jeremy N Kay |
| Pew Charitable Trusts | | Jeremy N Kay |
| McKnight Endowment Fund for Neuroscience | | Jeremy N Kay |
| Alfred P. Sloan Foundation | | Jeremy N Kay |

The funders had no role in study design, data collection and interpretation, or the decision to submit the work for publication.

### Author contributions

JJV, Protein production, purification and quantification; binding screen; primary literature research; microfluidic device fabrication; stripe assays; immunohistochemistry; manuscript review, Acquisition of data, Analysis and interpretation of data; YC, Protein production, purification and quantification; binding screen; primary literature research; microfluidic device design and fabrication; stripe assays; Acquisition of data; Analysis and interpretation of data; SCP, Protein production, purification and quantification; binding screen; primary literature research; immunohistochemistry; Acquisition of data; Analysis and interpretation of data; ABC, Cloning; protein production, purification and quantification; primary literature research; Acquisition of data; Analysis and interpretation of data; BP, Binding assay optimization; Acquisition of data; Analysis and interpretation of data; SSM, Primary literature research; microfluidic device fabrication; Acquisition of data; TAR, in situ hybridization and immunohistochemistry; Acquisition of data; JNK, Conception and design of research; microarray profiling; cloning; in situ hybridization and immunohistochemistry; manuscript review; Conception and design; Acquisition of data; Analysis and interpretation of data; Drafting or revising the article; WMW, Conception and design of research; bioinformatics and microarray analysis; binding assay optimization; cloning; protein production, purification and quantification; binding screen; primary literature research; cell aggregation assays and titration curves; stripe assays; immunohistochemistry;

prepared figures; wrote manuscript; Conception and design; Acquisition of data; Analysis and interpretation of data; Drafting or revising the article

### Ethics

Animal experimentation: All animal procedures were approved by the University of California, Berkeley (Office of Laboratory Animal Care (OLAC) protocol #R308) and they conformed to the National Institutes of Health Guide for the Care and Use of Laboratory Animals, the Public Health Service Policy and the Society for Neuroscience Policy on the Use of Animals in Neuroscience Research.

## Additional files

### Major datasets

The following datasets were generated:

| Author(s) | Year | Dataset title | Dataset URL | Database, license, and accessibility information |
|---|---|---|---|---|
| Visser JJ, Cheng Y, Perry SC, Chastain B, Parsa B, Masri SS, Kay JN, Wojtowicz WM | 2015 | Mouse retina extracellular receptor-ligand biochemical screen | http://dx.doi.org/10.5061/dryad.hf50r | Available at Dryad Digital Repository under a CC0 Public Domain Dedication |

The following previously published dataset was used:

| Author(s) | Year | Dataset title | Dataset URL | Database, license, and accessibility information |
|---|---|---|---|---|
| Kay JN, Sanes JR | 2012 | A gene expression database for retinal neuron subtypes | http://www.ncbi.nlm.nih.gov/geo/query/acc.cgi?acc=GSE35077 | Publicly available at the Gene Expression Omnibus (Accession no. GSE35077). |

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
