## [Decision Letter]

Thank you for submitting your work entitled "An extracellular biochemical screen identifies receptor-ligand interactions that direct neurite outgrowth in the retina" for peer review at *eLife*. Your submission has been favorably evaluated by a Senior editor, a Reviewing editor (Constance Cepko), and two reviewers, one of whom, Gavin Wright, has agreed to reveal his identity.

The reviewers have discussed the reviews with one another and the Reviewing editor has drafted this decision to help you prepare a revised submission.

Summary:

In its present form, this manuscript is preliminary and will need additional validation for acceptance in *eLife*. In particular, caution is urged in the assessment of the significance of interactions in the absence of a determination of the physiological relevance of identified binding partners; tetramerization of both ligand and receptor has the potential to generate binding interfaces that may not be representative of in vivo interactions. Further, pairwise interactions defined in vitro may not reflect the situation in vivo should additional proteins be part of the functional protein complex. Were the authors to address the following points to better support their molecular model for how SAC laminar specificity is generated, this work would be strengthened and could become a candidate for *eLife*.

Essential revisions:

1) The title of this manuscript claims to identify "receptor-ligand interactions that direct neurite outgrowth in the retina", however neurite outgrowth is not directly examined in this manuscript (outgrowth and lamination are not the same), and functional roles for FLRT2 and Unc5C not examined in vivo. Therefore, these claims appear overstated. The authors really need to examine IPL lamination in *Flrt2^-/-^*and *Unc5C^-/-^* mutant retinas, mutant mice that are available, to ask if FLRT+ SACs (see point 3) are repelled by Unc5C to modulate IPL lamination in vivo. It is recognized, however, that though these mutant mice are available, assessment of SAC lamination in these mutants likely falls outside of the revision timeframe *eLife* strives to achieve, and thus data from the KO mice are not required for the revision. However, if the authors have obtained these strains and have been able to investigate the phenotype, and the data validate the role suggested by the in vitro activities, it would greatly strengthen the paper and they are urged to submit these data.

2) The authors show that some retinal neurons are repelled by stripes of Unc5C, but they do not comment on the other Unc5 family members. Are the other Unc5's also capable of repelling retinal neurons?

3) By only examining in vivo expression of FLRT2 and Unc5C at P6, the conclusions the authors are able to draw regarding the possibility that the FLRT2-Unc5C interaction "mediates laminar organization of SAC neurons in the developing IPL via repulsion" are limited. To ask if this interaction is poised to guide lamination, as opposed to maintaining lamination established by other mechanisms, the authors should characterize the expression of FLRT2 and Unc5C at earlier developmental time points, when SACs and RGCs are first making decisions about neurite lamination neurites in the developing IPL. FLRT2 and Unc5C expression at P2 and P4, time points flanking SAC lamination in the IPL, should at the very least be examined. The authors reference the remaining FLRT and Unc5 expression patterns in the IPL as "data not shown" in the Discussion. These data should be included in the supplement to allow readers a comprehensive view of FLRT/Unc5 expression.

4) FLRT2 undergoes ectodomain shedding, which confounds the source of FLRT2 that colocalizes with VAChAT in the IPL. To show that FLRT2 is indeed expressed by SACs in vivo, the authors should show that FLRT2 expression colocalizes with ChAT+ SAC somas, and/or tdTomato expression in a *ChAT::cre;ROSA^LSL-tdTomato^*background in vivo, in addition to their in vitro analysis of FLRT2 expression in cultured SACs. Are FLRT2+ somas observable in the inner nuclear or ganglion cell layers? If this is not so at P6, what about at earlier time points? Does RNA in situ analysis allow for a clearer picture of which cell times express these proteins, and when?

5) To support the idea that Unc5C initiates FLRT2-mediated repulsive signaling in retinal neurons, the authors could capitalize on the specificity of the anti-FLRT2 reagent they use here to carry out function-blocking experiments in their stripe assay experiments. If retinal neurons cultured in the presence of anti-FLRT2 no longer avoid Unc5C stripes, and vice versa, this would strengthen the model that FLRT2 is indeed a receptor for repulsive signaling via Unc5C (and vice versa), as opposed to some other unidentified receptor for Unc5C. It is especially important to show that FLRT2 mediates the Unc5C repulsion, since the ectodomain of FLRTs can be shed.

6) The repulsive interactions observed from total dissociated retinas involve ~10-15% of total retinal neurites in these assays. Though SACs are subsequently investigated in this study, the double immunolabeling for distinct classes of retinal subtypes should be employed in these stripe assays to determine which classes of retinal neurons are indeed repelled by UNC5-are they all SACs, or do other subtypes similarly show these repulsive interactions?

7) The main contribution of this manuscript will be the novel interactions they have identified, but they really didn't make much of these novel interactions in the manuscript at all; rather, they focussed on providing a thorough analysis of known interactions e.g. the Sema-Plexin (Table 1) and Flrt-Unc5 (Figure 2) interactomes. The manuscript can be improved by a brief description/discussion of some novel interactions.

8) In Figure 4 please explain why it was necessary to use antibodies independently on separate sections rather than double labelling on a single section. The finding that the expression is complementary is convincing, but it would be more so if double labelling were performed.

---

## [Author Response]

Essential revisions:

1) The title of this manuscript claims to identify "receptor-ligand interactions that direct neurite outgrowth in the retina", however neurite outgrowth is not directly examined in this manuscript (outgrowth and lamination are not the same), and functional roles for FLRT2 and Unc5C not examined in vivo. Therefore, these claims appear overstated.

We understand and agree with the reviewers’ points that neurite outgrowth and lamination are not the same and that we have not examined functional roles for FLRT2 and Unc5C in vivo. As such, we have chosen a new title. In light of the additional expression and stripe assay data we have obtained for the entire FLRT and Unc5 families (described in detail below), we have revised the title to: "An extracellular biochemical screen reveals that FLRTs and Unc5s mediate neuronal subtype recognition in the developing retina". As our data reflect responses of primary retinal neurons to FLRT and Unc5 proteins ex vivo, we believe it is most accurate to state very generally in the title that these proteins can mediate "recognition" (as evidenced by attraction and repulsion in stripe assays) and we will wait for future functional data to make any statements regarding a definitive role for these proteins in lamination.

*The authors really need to examine IPL lamination in Flrt2^-/-^and Unc5C^-/-^ mutant retinas, mutant mice that are available, to ask if FLRT+ SACs (see point 3) are repelled by Unc5C to modulate IPL lamination in vivo. It is recognized, however, that though these mutant mice are available, assessment of SAC lamination in these mutants likely falls outside of the revision timeframe* eLife *strives to achieve, and thus data from the KO mice are not required for the revision. However, if the authors have obtained these strains and have been able to investigate the phenotype, and the data validate the role suggested by the in vitro activities, it would greatly strengthen the paper and they are urged to submit these data.*

We discussed the KO mouse experiments with the editor who, after consulting with the reviewers, decided that, since we had not obtained the *Flrt2^-/-^*and *Unc5C^-/-^*strains, these experiments would not be required for acceptance of our manuscript.

2) The authors show that some retinal neurons are repelled by stripes of Unc5C, but they do not comment on the other Unc5 family members. Are the other Unc5's also capable of repelling retinal neurons?

To address this important point, we decided to conduct a thorough analysis of all FLRTs and Unc5s and performed stripe assays using all proteins. We found that most members of these protein families elicit responses from subpopulations of neurons (5-18%) in primary retina cultures. Three proteins were exclusively repulsive (FLRT2, Unc5C, Unc5D) while two were capable of attracting some retinal neurons and repelling others (FLRT1, FLRT3). Only Unc5B (which is barely expressed in the developing retina; see Figure 5) and Unc5A were incapable of eliciting responses. Example images of the effect of the FLRTs and Unc5s on neuron outgrowth in culture and quantification of the data are included in Figure 4.

3) By only examining in vivo expression of FLRT2 and Unc5C at P6, the conclusions the authors are able to draw regarding the possibility that the FLRT2-Unc5C interaction "mediates laminar organization of SAC neurons in the developing IPL via repulsion" are limited. To ask if this interaction is poised to guide lamination, as opposed to maintaining lamination established by other mechanisms, the authors should characterize the expression of FLRT2 and Unc5C at earlier developmental time points, when SACs and RGCs are first making decisions about neurite lamination neurites in the developing IPL. FLRT2 and Unc5C expression at P2 and P4, time points flanking SAC lamination in the IPL, should at the very least be examined. The authors reference the remaining FLRT and Unc5 expression patterns in the IPL as "data not shown" in the Discussion. These data should be included in the supplement to allow readers a comprehensive view of FLRT/Unc5 expression.

We conducted immunostaining for all FLRTs and Unc5s at P2, P4 and P6 and now present a thorough developmental analysis of all family members (Figure 5 and Figure 5—figure supplement 1). For FLRT2 and Unc5C, we found that they both exhibit diffuse expression across the IPL at P2 but, by P4, exhibit laminar restriction in S2/4 and S1/3/5, respectively. As such, expression analyses demonstrate that these proteins are expressed at the right time and place to play a role in mediating and/or maintaining SAC lamination. Moreover, this analysis showed that only FLRT2 is specifically expressed in the DS circuit IPL sublayers S2/4 (i.e., none of the other FLRTs or Unc5s exhibit S2/4 laminar restriction), supporting our observations from immunostaining dissociated ChatCre neurons with antibodies against the other FLRTs and all of the Unc5s (Figure 6 and Figure 6—figure supplement 1). In addition, by performing P2, P4 and P6 developmental analyses of all members of the FLRTs and Unc5s families by immunostaining, we found that each protein (with the exception of Unc5B) exhibits a unique pattern of lamina-specific expression, suggesting that these proteins may play a role in mediating recognition events that direct laminar specificity.

4) FLRT2 undergoes ectodomain shedding, which confounds the source of FLRT2 that colocalizes with VAChAT in the IPL. To show that FLRT2 is indeed expressed by SACs in vivo, the authors should show that FLRT2 expression colocalizes with ChAT+ SAC somas, and/or tdTomato expression in a ChAT::cre;ROSA^LSL-tdTomato^background in vivo, in addition to their in vitro analysis of FLRT2 expression in cultured SACs. Are FLRT2+ somas observable in the inner nuclear or ganglion cell layers? If this is not so at P6, what about at earlier time points? Does RNA in situ analysis allow for a clearer picture of which cell times express these proteins, and when?

The reviewers requested analysis in vivo. As FLRT2+ somas are not observable in the INL or GCL at P6 (data from our previous submission), we performed FLRT2 immunostaining at P2 and P4 but observed no soma staining at these earlier points either (Figure 5—figure supplement 1). As such, to obtain cell body staining, we performed *Flrt2* in situhybridization. We found that, at both P1 and P6, *Flrt2* is expressed in a subpopulation of cells that include: 1) SACs; 2) a small subset of non-SAC cells in the INL; and 3) a small subset of non-SACs in the GCL. This latter group was further shown to include ooDSGCs, another cell type that projects to S2/4. These data are presented in Figure 6. Together, these analyses demonstrate that two cell types that project to S2 and S4 (i.e. SACs and ooDSGCs) express the *Flrt2* gene (based on in situhybridization) and FLRT2 protein (based on immunostaining of dissociated neurons).

5) To support the idea that Unc5C initiates FLRT2-mediated repulsive signaling in retinal neurons, the authors could capitalize on the specificity of the anti-FLRT2 reagent they use here to carry out function-blocking experiments in their stripe assay experiments. If retinal neurons cultured in the presence of anti-FLRT2 no longer avoid Unc5C stripes, and vice versa, this would strengthen the model that FLRT2 is indeed a receptor for repulsive signaling via Unc5C (and vice versa), as opposed to some other unidentified receptor for Unc5C. It is especially important to show that FLRT2 mediates the Unc5C repulsion, since the ectodomain of FLRTs can be shed.

We agree that this loss-of-function stripe assay is a terrific experiment and we pushed really hard on it prior to the initial submission of our manuscript. Unfortunately, even with adding fresh antibody to the media every 24 hours (a prohibitively expensive experiment), we were unable to detect any decrease in repulsion. Regrettably, this negative result does not allow us to distinguish between the possibility that the experiment failed for technical reasons and the possibility that blocking FLRT2 has no effect. Prior to resubmission we discussed this experiment with the editor who, after consulting with the reviewers, decided that these experiments would not be required for acceptance of our manuscript.

As an alternative route to addressing the question of whether FLRT2 can mediate repulsion, we sought to ask whether misexpression of FLRT2 can render retinal neurons sensitive to Unc5C stripes. Since only 11% of wild type neurons are repelled by Unc5C stripes (Figure 4) and *Flrt2* RNA is expressed by only a small subset of retinal neurons (Figure 6), we reasoned that a gain-of-function approach might be feasible. Our first step was to find a transfection reagent and conditions that would allow efficient transfection of primary retinal neurons, something we had tried prior to our initial submission but were never successful in achieving. We pushed really hard, testing many different commercially-available transfection reagents and conditions and, while most were incapable of producing transfected neurons in our primary retinal neuron cultures, we eventually found one that gave us a 10% transfection efficiency making it possible to perform this experiment. However, while we could easily get GFP transfected neurons in our cultures (this is how we tested the various transfection reagents), we were unable to get FLRT-myc expressed (this was a problem specific to our neuronal cultures as FLRT2-myc expressed well in CHO.K1 cells). In the end, moving our FLRT2-myc transgene into a different expression vector (ironically, pUb from the Cepko lab) solved the problem (!) and we were able to get transgene expression and perform the gain-of-function experiment.

We found that neurons ectopically expressing FLRT2-myc were repelled by Unc5C stripes while control- transfected neurons were not. As such, FLRT2 expression in primary retinal neurons is sufficient to confer repulsion in response to Unc5C. This finding, when taken together with our data demonstrating that FLRT2 binds directly to Unc5C in our biochemical assays, strongly suggests that transcellular FLRT2- Unc5C interactions initiate FLRT2-mediated repulsion. At the very least, we can conclude that FLRT2 expression assembles a repulsive signaling complex that can respond to extracellular Unc5C.

6) The repulsive interactions observed from total dissociated retinas involve ~10-15% of total retinal neurites in these assays. Though SACs are subsequently investigated in this study, the double immunolabeling for distinct classes of retinal subtypes should be employed in these stripe assays to determine which classes of retinal neurons are indeed repelled by UNC5-are they all SACs, or do other subtypes similarly show these repulsive interactions?

As the reviewers point out, in our initial submission we reported that we observed non-SAC, FLRT2+ neurons that are repelled by Unc5C in our stripe assays. To investigate the identity of these non-SAC neurons, we performed in situhybridization for *Flrt2* at P1 and P6 and observed that a small number of non-SAC cells are *Flrt2*+: 1) non-SACs in the INL and 2) non-SACs in the GCL. Based upon the large soma size of the *Flrt2*+ cells in the GCL, and the fact that DSGCs arborize in S2/4 and are post-synaptic partners of SACs, we investigated whether DSGCs might be part of the group of non-SAC neurons that are repelled by Unc5C. First, we found by double in situhybridization and antibody staining that *Flrt2+* GCL cells express CART, a marker of ooDSGCs. Using neurons harvested from Drd4-GFP mice (which label a subset of ooDSGCs), we found that Drd4-GFP neurons express FLRT2 protein and are repelled by Unc5C stripes. Thus, two members of the DS circuit, SACs and Drd4-GFP+ ooDSGCs exhibit identical growth responses to Unc5C in stripe assays. We were unable to determine the identity of the non-SAC cells in the INL, although their laminar position close to the IPL indicates they are an amacrine subtype. Furthermore, the fact that these cells are present at P1 further supports this conclusion, as bipolar cells are not yet born at P1.

7) The main contribution of this manuscript will be the novel interactions they have identified, but they really didn't make much of these novel interactions in the manuscript at all; rather, they focussed on providing a thorough analysis of known interactions e.g. the Sema-Plexin (Table 1) and Flrt-Unc5 (Figure 2) interactomes. The manuscript can be improved by a brief description/discussion of some novel interactions.

We thank the reviewers for their kind words – we agree that this is an important contribution of our work, and that we did not include enough details in the first submission for readers to appreciate this. We have included a more thorough discussion of novel interactions in the Results and have added a main text figure describing selected interactions that we expect will be particularly interesting to the field.

8) In Figure 4 please explain why it was necessary to use antibodies independently on separate sections rather than double labelling on a single section. The finding that the expression is complementary is convincing, but it would be more so if double labelling were performed.

We agree that double-labeling with FLRT2 and Unc5C is ideal. However, this experiment is not technically possible because both the Unc5C and the FLRT2 antibodies are rabbit. As such, we needed to show complementary expression using the S2/4 marker, VAChT. Note that in the Methods of our initial submission, the Unc5C antibody was erroneously listed as goat. This typo likely arose due to the fact that we do indeed have a goat Unc5C antibody in the lab but we did not use it for immunostaining because, upon initial biochemical characterization, it showed cross-reactivity with other Unc5s in ELISA experiments.